# Multi-User Dueling Bandits:
# A Fair Approach using Nash Social Welfare

## Abstract

Learning from human preference data is becoming a useful tool, from fine-tuning large language models to training reinforcement learning agents. However, in most scenarios, the model is trained on the average preference of all human evaluators, which, under large variations of preferences, can be unfair to minority groups. In this work, we consider fairness in dueling bandits, a standard framework for online learning from preference data. We assume that each user has a (potentially distinct) Condorcet winner, which is an arm preferred to every other arm. Using these user-specific Condorcet winners as reference points, we evaluate and score arms according to their performance relative to the corresponding winner. To promote fairness across heterogeneous users, we adopt the well-established Nash Social Welfare objective, which maximizes the product of user utilities, thereby inherently penalizing inequality and preventing the marginalization of any single user. Within this framework, we construct a hard instance to establish a regret lower bound of $\Omega(T^{2/3}\min(K, D)^{\frac{1}{3}})$ for a time horizon $T$, $K$ arms, and $D$ users, which, to the best of our knowledge, is the first result quantifying the cost of fairness in dueling bandits with heterogeneous preferences. We then present the Fair-Explore-Then-Commit and Fair-$\epsilon$-Greedy algorithms with a Condorcet winner identification phase. We further derive their regret upper bounds that match the lower-bound dependence on $T$ up to logarithmic factors.

## 1 Introduction

The paradigm of learning from human feedback has become central to the development of modern machine learning algorithms, enabling models to align with complex, often non-scalar human values and individual preferences. This is evident in applications ranging from the fine-tuning of large language models (Ziegler et al., 2019) to the training of reinforcement learning agents (Christiano et al., 2017). However, a significant challenge emerges when this feedback is aggregated from a diverse group of users, each with their own unique preferences. A common approach is to optimize for the average preference, a strategy that, while simple, can inadvertently lead to policies that favor the statistical majority at the expense of minority groups whose opinions diverge from the norm. This can result in a system that is perceived as unfair or biased, undermining the very goal of human-centric machine learning.

The dueling bandit problem provides a formal and well-suited framework for this type of online learning from pairwise preference data. Unlike traditional multi-armed bandits, where a single, scalar reward is received for each action, dueling bandits receive a relative preference signal, which is a binary outcome indicating which of two presented options is preferred. This makes the model particularly relevant for applications like information retrieval, recommender systems, and personalized medicine, where it is often easier for a human to compare two items than to provide a numerical rating for a single one (Yue et al., 2012).

In the traditional dueling bandit setting, the agent is usually trained using data coming from a single preference model, often corresponding to a single human or the average preference of multiple persons. However, in practice, preferences can be collected from a set of diverse preference models that can be inferred from users' behavior, such as explicit pairwise choices, clicks, skips, ratings, or purchases. For example, a restaurant's *discounted dish of the week* is a single promotional decision affecting multiple users. Different

customer segments may consistently prefer different dishes, but if the system aggregates inferred feedback and keeps promoting from the options that look best on average, minority tastes will be underserved in the long run. Likewise, consider background music in a shared workplace or public venue where the playlist or genre is chosen daily. Optimizing for aggregate engagement can chronically expose some users to music they dislike while privileging the preferences of the largest group.

Motivated by this example, we look at settings where a single decision maker takes actions that affect multiple users. Our work addresses this challenge by introducing fairness considerations into the dueling bandit framework. We propose a novel formulation where multiple users with potentially conflicting preference matrices provide feedback, and the goal is to learn policies that fairly balance their interests. Drawing from the rich literature on social choice theory and welfare economics (Nash, 1950; Kaneko & Nakamura, 1979; Moulin, 2004), we adopt the Nash Social Welfare (NSW) function as our fairness criterion.

Our work is inspired by recent research on fairness in bandits and RL. For instance, Hossain et al. (2021) introduced a multi-agent multi-armed bandit model and used NSW as the fairness criterion to design algorithms with sublinear regret. Jones et al. (2023) and Zhang et al. (2024) further study multi-agent multi-armed bandits under NSW, providing improved algorithms and regret bounds. However, these works operate in the standard multi-armed bandit framework where agents observe absolute rewards. In contrast, our work addresses the *dueling bandit* setting where only relative pairwise preferences are available, requiring a fundamentally different approach to utility estimation and algorithm design.

**Contributions.** Our contributions can be summarized as follows:

- **Problem Formulation:** We formalize the *Fair Multi-User Dueling Bandit* problem. Unlike standard settings that aggregate feedback into a single preference matrix, we model distinct user utilities based on their individual Condorcet winners and adopt the *Nash Social Welfare* as a fairness objective to balance utility across users.

- **Regret Lower Bound:** We identify and construct hard instances of the formulated fair multi-user dueling bandit problem. By analyzing these instances, we prove the fundamental hardness of this setting and establish a regret lower bound of $\Omega(T^{2/3} \min(K, D)^{\frac{1}{3}})$, where $T$ is the training horizon.

- **Dueling Bandit Algorithms with Theoretical Regret Upper Bounds:** We propose two algorithms, Fair-Explore-Then-Commit and Fair-$\epsilon$-Greedy, adapted for this multi-user setting. We provide theoretical guarantees showing that both algorithms achieve a regret of $\tilde{O}(C(\phi)T^{2/3})$, where $C(\phi)$ is a function of the instance parameters such as the number of arms. The upper bound matches the lower bound with respect to the time horizon $T$ (up to logarithmic factors).

## 2 Related Work

Given the critical importance of fairness in machine learning, a significant body of research has been dedicated to defining and addressing algorithmic bias across various domains (Dwork et al., 2012; Hardt et al., 2016; Kleinberg et al., 2016; Kusner et al., 2017). In this section, we focus specifically on the subset of this literature that addresses fairness within multi-armed bandits and reinforcement learning.

**Fairness in Multi-armed Bandits** The intersection of fairness and bandit algorithms has received considerable attention in recent years. Hossain et al. (2021) introduced a multi-agent variant of the classical multi-armed bandit problem where multiple agents receive potentially different rewards from each arm, and the goal is to learn a fair distribution over arms using the Nash social welfare as the fairness criterion. They design three algorithms for the problem setup: explore-then-commit, $\epsilon$-greedy, and upper confidence bound (UCB), and prove that these algorithms achieve sublinear regret with UCB's regret being $\tilde{O}(\min(DK, \sqrt{D}K^{\frac{3}{2}})\sqrt{T})$. Jones et al. (2023) continue this line of work, addressing the computational inefficiency of prior methods. They propose a computationally efficient algorithm with an improved regret of $\tilde{O}(\sqrt{DKT} + DK)$. Additionally, Zhang et al. (2024) analyzed the problem under the strict geometric mean definition of the NSW function, compared to the product of utility definition used in prior work. They

established tight regret upper bounds of $O(K^{\frac{2}{D}}T^{\frac{D-1}{D}} + K)$ for the stochastic setting while proving that sublinear regret is achievable in the adversarial setting only under specific approximations, as the strict case is impossible. While the aforementioned works focus on fairness among multiple users at each step, another line of research evaluates fairness across the horizon by defining the Nash Social Welfare over the rewards accumulated across all rounds $T$, leading to the study of Nash regret (Barman et al., 2023; Sawarni et al., 2023). Crucially, this setting emphasizes that fairness is achieved across distinct user encounters over time, as the agent is assumed to interact with different users at each individual time step.

**Fairness in Reinforcement Learning** Our work also relates to recent efforts on fairness in multi-agent and multi-objective RL. Siddique et al. (2020) explored Nash welfare as a fairness metric by proposing Q-learning variants that maximize NSW of accumulated rewards across multiple objectives. Mandal & Gan (2022) take an axiomatic approach, showing that among common fairness objectives such as minimum welfare and generalized Gini social welfare, the Nash social welfare uniquely satisfies all natural axioms in multi-agent RL. They further propose RL algorithms with regret guarantees for the NSW objective in episodic MDPs. Fan et al. (2022) similarly consider multi-objective RL, framing it as a welfare-maximization problem. They develop Q-learning methods that converge to policies optimizing expected NSW compared to the NSW of the expected rewards in prior work. Kim et al. (2025) consider the generalized $p$-means welfare functions, which include the Nash social welfare function. They design an algorithm that outputs a set of near-optimal policies for $p \leq 1$.

Overall, the use of the Nash social welfare objective for fairness has been justified both axiomatically and empirically across different settings. Our contribution is to bring this perspective into the dueling bandit setting with multiple users, where preference-based feedback requires new algorithmic techniques and analysis.

## 3 Preliminaries

In this section, we formally state the fair multi-user dueling bandit problem. We introduce the core components of our framework, including the multi-user preference model, the scoring functions derived from Condorcet winners, and the Nash social welfare objective. These definitions establish the foundation for the algorithmic solutions and theoretical analysis presented in subsequent sections.

### 3.1 Problem Definition

Consider a multi-armed dueling bandit setting where an agent has access to $K$ arms, and the actions are evaluated by $D$ users. At each time step $t$, the agent chooses two actions $a_t$ and $a'_t$. Actions $a_t$ and $a'_t$ are sampled from policies $\pi_t, \pi'_t \in \Delta^K$ with probability $\pi_t(a)$ and $\pi'_t(a')$ simultaneously where $\Delta^K$ is the probability simplex of dimension $K$.

At each step, the agent observes a binary feedback vector $\mathbf{y}_t \in \{0, 1\}^D$, where the $d$-th entry $y_t[d]$ indicates whether user $d$ preferred $a_t$ over $a'_t$ (value 1) or vice versa (value 0). The underlying preferences for each user $d$ are governed by a matrix $\mathcal{P}_d \in [0, 1]^{K \times K}$, where the entry $\mathcal{P}_{d,i,j}$ represents the probability that user $d$ prefers arm $i$ over arm $j$. Collectively, these matrices form the preference tensor $\mathcal{P} \in [0, 1]^{D \times K \times K}$. The preference matrices satisfy the standard reciprocal property, such that for any user $d$ and pair of arms $(i, j)$, $\mathcal{P}_{d,i,j} + \mathcal{P}_{d,j,i} = 1$ and $\mathcal{P}_{d,i,i} = 0.5$.

While the preference tensor $\mathcal{P}$ captures the relative strength of arms in pairwise comparisons, our goal of maximizing social welfare requires a measure of an arm's utility for each user. To bridge this gap, we introduce a scoring function $s : [K] \times [0, 1]^{D \times K \times K} \to [0, 1]^{D \times K}$. Intuitively, this function maps the pairwise preference relations into a scalar score representing the quality of an arm for a specific user. Formally, it assigns a vector of scores $s(a, \mathcal{P}) \in [0, 1]^D$ to an arm $a$, which we denote as $s_d(a)$ for user $d$, serving as the fundamental unit of utility for our objective.

We define the global objective function $f : \Delta^K \times [0, 1]^{D \times K} \to \mathbb{R}$ to quantify the total social welfare of a policy by aggregating individual user utilities (derived from the scores $s$). This function serves as the maximization target for the agent. Following the standard dueling bandit literature, we define the instantaneous regret $r_t$ as the performance gap between the optimal policy $\pi^*$ and the agent's policies at time $t$. Since the agent

selects two policies $\pi_t$ and $\pi'_t$ at each step, the regret is calculated as the difference between the optimal value and the average welfare of the selected policies. The instantaneous regret at time t is defined as

$$r_t = f(\pi^*, s) - \frac{f(\pi_t, s) + f(\pi'_t, s)}{2},$$

where $\pi^* \in \arg\max f(\pi, s)$ is an optimal policy and $\pi_t$ and $\pi'_t$ are the agent's policies at time step $t$. The goal is to design an algorithm that minimizes the expected total regret, defined as

$$\mathbb{E}[R_T] = \mathbb{E}\left[\sum_{t=1}^{T} r_t\right] = \mathbb{E}\left[\sum_{t=1}^{T} f(\pi^*, s) - \frac{f(\pi_t, s) + f(\pi'_t, s)}{2}\right], \tag{1}$$

throughout the learning process until horizon $T$.

### 3.2 Condorcet Winner

In the dueling bandit framework, feedback is relative, lacking the absolute rewards typically used to calculate welfare. To establish a rigorous measure of user utility, which is essential for optimizing a Social Welfare function, we require a personalized reference point for each user. We adopt the Condorcet winner as this benchmark. Formally, a Condorcet winner for user $d$ is defined as an arm $a_d^* \in [K]$ such that $P_{d,a_d^*,j} > 0.5$, $\forall j \in [K], j \neq a_d^*$. By definition, such an arm is unique if it exists. While the Condorcet winner is not guaranteed to exist, its existence is a common assumption in the literature (Urvoy et al., 2013; Zoghi et al., 2014; Komiyama et al., 2015; Chen & Frazier, 2017; Xu et al., 2019; Haddenhorst et al., 2021). This assumption implies that for each user, there exists an arm that wins against any other arm with probability greater than 0.5. We assume the existence of a Condorcet winner for each user which we state formally in Assumption 3.1.

**Assumption 3.1.** For every user $d \in [D]$, there exists a unique arm $a_d^* \in [K]$ such that $\mathcal{P}_d(a_d^* \succ j) > 0.5$ for all $j \neq a_d^*$.

Leveraging the Condorcet winner as a reference point, we define the scoring function to quantify the performance of an arm relative to each user's Condorcet winner. We define the score of arm $i \in [K]$ for user $d \in [D]$ as the scaled probability that arm $i$ beats the Condorcet winner $a_d^*$ which is formally defined as

$$s_d(i) = 2\mathcal{P}_{d,i,a_d^*}. \tag{2}$$

This scaling effectively normalizes the utility to the $[0, 1]$ interval. It guarantees a maximum score of 1 (since $s_d(a_d^*) = 2\mathcal{P}_{d,a_d^*,a_d^*} = 2 \times 0.5 = 1$) and the lower bound is 0 according to the definition of $\mathcal{P}_d$.

### 3.3 Nash Social Welfare

To quantify fairness in our setting, we adopt the Nash Social Welfare (NSW) as our central objective function. Originating in game theory, this concept is designed to balance efficient and fair resource allocation among entities, ensuring that no single user's utility is ignored. Given the set $A$ of all possible allocations and the utility function $u_d : A \to \mathbb{R}^+$ for each user $d \in D$, the NSW function is defined as

$$NSW(a) = \prod_{d=1}^{D} u_d(a),$$

where $a \in A$ is an allocation. The NSW function is the unique function that satisfies four axioms: Pareto optimality, independence of irrelevant alternatives, anonymity, and continuity (Kaneko & Nakamura, 1979). Moreover, the NSW function was shown to satisfy some fairness metrics, such as envy freeness up to one good and proportionality (McGlaughlin & Garg, 2020; Caragiannis et al., 2019), and has been extensively used as a metric for fairness in different settings (Brânzei et al., 2017; Liao et al., 2024; Jagtenberg & Mason, 2020).

Most notably, Hossain et al. (2021) uses the NSW function as a fairness metric to learn a fair distribution over arms in a multi-agent multi-armed bandit setting.

Adapting this to our dueling bandit framework, we define a user's utility function for a policy as the expected score, i.e., $u_d(\pi) = \mathbb{E}_{a \sim \pi}[s_d(a)]$. Hence, our specific social welfare function $f$ is given by:

$$f(\pi, s) = NSW(\pi, s) = \prod_{d=1}^{D} \left( \mathbb{E}_{a \sim \pi}[s_d(a)] \right), \tag{3}$$

and the total regret can be defined as

$$\mathbb{E}[R_T] = \mathbb{E}\left[ \sum_{t=1}^{T} r_t \right] = \mathbb{E}\left[ \sum_{t=1}^{T} \prod_{d=1}^{D} \mathbb{E}_{a \sim \pi^*}[2\mathcal{P}_{d,a,a_d^*}] - \frac{\prod_{d=1}^{D} \mathbb{E}_{a \sim \pi_t}[2\mathcal{P}_{d,a,a_d^*}] + \prod_{d=1}^{D} \mathbb{E}_{a \sim \pi_t'}[2\mathcal{P}_{d,a,a_d^*}]}{2} \right]. \tag{4}$$

Before proceeding, it is important to clarify the connection between the product formulation of the Nash Social Welfare and its geometric mean representation. Since $x \to x^{1/D}$ is strictly increasing on $[0, \infty)$, the product $\prod_{d=1}^{D} u_d(\pi)$ and its $D$-th root, the geometric mean $\left( \prod_{d=1}^{D} u_d(\pi) \right)^{1/D}$, induce the same ordering over policies $\pi$: a policy maximizes one if and only if it maximizes the other. Hence, any policy optimal for one is optimal for the other.

## 4 Fair Multi-User Dueling Bandit

In this section, we present our main theoretical contributions for the fair multi-user dueling bandit problem. We begin by establishing the fundamental hardness of the setting, deriving a regret lower bound. Following this analysis, we introduce two algorithms designed to efficiently optimize the NSW objective: the *Fair-Explore-Then-Commit* and the *Fair-$\epsilon$-Greedy* algorithms. We provide a rigorous theoretical analysis for both methods, establishing regret upper bounds that match the lower bound dependence on horizon $T$ up to logarithmic factors.

### 4.1 Hardness Result

We now turn to the fundamental information-theoretic limits of the fair multi-user dueling bandit problem. While standard dueling bandit algorithms can achieve $O(\log T)$ regret (Zoghi et al., 2014), we show that the requirement to maximize the NSW among users with conflicting Condorcet winners introduces significant additional hardness.

The core difficulty arises from the need to play the Condorcet winners of users in order to estimate the scores of arms. Consider user $d$ with the Condorcet winner $a_d^*$ with $s_{d'}(a_d^*) = 0$, $\forall d' \neq d$. While the arm $a_d^*$ is a Condorcet winner for user $d$, playing the arm yields zero score to all other users. However, in order to estimate the scores $s_d(a)$ for $a \neq a_d^*$, the pair $(a, a_d^*)$ needs to be played. We utilize this property to create a set of hard instances and derive the lower bound in Theorem 4.1.

**Theorem 4.1.** *Consider the fair multi-user dueling bandit problem with $D$ users and $K$ arms, where $D, K \geq 4$, and a time horizon $T$. Suppose the user preferences satisfy Assumption 3.1 and the utilities are defined by the scoring function in Equation 2. Then, for any online learning algorithm, there exists a problem instance such that the expected regret (defined in Equation 4) is at least:*

$$\mathbb{E}[R_T] = \Omega\left( T^{2/3} \min(K, D)^{1/3} \right).$$

*Proof Sketch of Theorem 4.1.* **Intuition:** The core difficulty lies in optimizing the NSW function when users have conflicting interests. We construct a needle in a haystack scenario where the algorithm must identify a subset of *Good* Condorcet winners among many *Bad* ones.

Consider the set of Condorcet winners of all users. We categorize them into *Bad* winners and *Good* winners. A *Bad Winner* is optimal for a specific user but yields zero (or very low) utility for all other users. Playing

---

**Algorithm 1** Fair-ETC

---
**Require:** Horizon $T$, Set of Arms $[K]$, Set of Users $[D]$, sample access to the joint preference tensor $\mathcal{P}$, $\delta$ confidence for DKWT, $L$ exploration steps.
1: $(\hat{A}^*, t) \leftarrow \mathbf{DKWT}(K, D, \mathcal{P}, \frac{\delta}{D})$.
2: Initialize score estimates $\hat{\mathcal{P}}_d(a, a_d^*) \leftarrow 0$ for all $a \in [K], d \in [D]$.
3: **for** $a^* \in \hat{A}^*$, $a \in [K]$, $i \in [L]$ **do**
4:     Play pair $(a, a^*)$ and observe feedback vector $\mathbf{y}_t$.
5:     Update $\hat{\mathcal{P}}_d(a, a^*) \leftarrow \hat{\mathcal{P}}_d(a, a^*) + \frac{1}{L}\mathbb{I}(\mathbf{y}_t[d] = 1)$ for all $d \in [D]$.
6:     $t \leftarrow t + 1$.
7: **end for**
8: Compute $\hat{\pi} \in \arg\max_{\pi \in \Delta^K} \prod_{d=1}^{D} \mathbb{E}_{a \sim \pi}[2\hat{\mathcal{P}}_d(a, a_d^*)]$
9: **while** $t \leq T$ **do**
10:     Sample arms $a_t, a_t'$ independently from $\hat{\pi}$
11:     $t \leftarrow t + 1$.
12: **end while**

---

a bad winner essentially zeroes out the NSW product. A *Good Winner*, while still optimal for its specific user, provides non-zero utility to others. The algorithm must distinguish between these two types to avoid incurring linear regret in the time horizon $T$ with respect to the NSW objective. Among these *Good Winners*, there exists one arm that is optimal. The statistical hardness comes from the fact that the preference margins distinguishing these instances are controlled by a small perturbation $\epsilon$.

**Hard Instance Construction:** We consider $D$ users and $K$ arms. We denote the set of Condorcet winners as $\mathcal{A}^*$, where each user $d$ has a specific winner $a_d^*$. The set $\mathcal{A}^*$ is partitioned into two sets, *Good Winners* ($\bar{\mathcal{A}}$) and *Bad Winners* ($\mathcal{A}^* \setminus \bar{\mathcal{A}}$) of size $|\mathcal{A}^*|/2$ each. A *Good Winner* $a_d^* \in \bar{\mathcal{A}}$ has scores that are close to 1 for other users, while a *Bad Winner* $a_{d'}^* \in \mathcal{A}^* \setminus \bar{\mathcal{A}}$ has a zero score for other users.

We construct $|\bar{\mathcal{A}}|$ distinct problem instances. In the $m$-th instance $\mathcal{I}^m$, we perturb the preferences such that a policy that plays arm $m \in \bar{\mathcal{A}}$ deterministically is optimal. The critical difference between arm $m$ and another good winner is their probability of winning against each bad winner according to the bad winner's corresponding users. Specifically, $\mathcal{P}_d(m, a_d^*) - \mathcal{P}_d(a_{d'}^*, a_d^*) = \epsilon, \forall a_{d'}^* \in \bar{\mathcal{A}}, a_{d'}^* \neq m$, and $a_d^* \in \mathcal{A}^* \setminus \bar{\mathcal{A}}$.

**Regret Analysis:** In instance $\mathcal{I}^m$, if the algorithm fails to identify the optimal arm $m$, it will play arms from $\bar{\mathcal{A}}$ with an $\epsilon$ sub-optimality. On the other hand, the agent needs to play the bad winners in order to distinguish arm $m$ from the good winners.

Consequently, the algorithm faces a fundamental dilemma. To distinguish the optimal arm $m$ from the other statistically similar good winners, it must play policies that select bad winners, which incurs high instantaneous regret. Conversely, if the algorithm avoids these bad winners, it fails to identify $m$ and suffers regret proportional to the sub-optimality gap $\epsilon$ over the horizon $T$. Balancing the regret from insufficient information against the cost of exploration leads to a worst-case instance where the perturbation is set to $\epsilon = \Theta\left(\left(\frac{|\mathcal{A}^*|}{T}\right)^{1/3}\right)$, resulting in the overall lower bound of $\Omega(T^{2/3}|\mathcal{A}^*|^{\frac{1}{3}})$. By observing that the set of Condorcet winners $\mathcal{A}^*$ is constrained by both the number of arms and users such that $|\mathcal{A}^*| \leq \min(K, D)$, and that in our constructed instances where this bound is tight, we obtain the final lower bound of $\Omega(T^{2/3}\min(K, D)^{1/3})$. □

### 4.2 Fair-Explore-then-Commit Algorithm

We design the Fair-Explore-Then-Commit (Fair-ETC) algorithm for fair multi-user dueling bandits and present it in Algorithm 1. The algorithm consists of three phases: (1) the Condorcet winner identification phase, (2) the exploration phase, where the scores of arms are estimated, and (3) the exploitation phase, where the agent plays the optimal policy that maximizes the NSW based on the estimated scores.

To identify the set of Condorcet winners $\hat{A}^* = \{\hat{a}_1^*, \ldots, \hat{a}_D^*\}$, we employ a modified version of the Dvoretzky-Kiefer-Wolfowitz Tournament (DKWT) algorithm (Haddenhorst et al., 2021). The standard DKWT was

introduced as a sample-efficient solution for identifying the generalized Condorcet winner in multi-dueling bandit scenarios. It operates as a round-based elimination procedure that iteratively discards arms from a candidate set once it can be proven, with high confidence, that they are not the winner.

At the core of this algorithm is a sequential mode-estimation subroutine built upon the Dvoretzky-Kiefer-Wolfowitz (DKW) inequality. The DKW inequality provides a tight, uniform bound on the maximum deviation between empirical and true categorical probability distributions. Rather than forcing a premature choice, the DKWT subroutine is designed to update the required confidence probability and continue exploration, if the empirical margins between competing arms do not yet satisfy the required confidence. By sequentially refining these confidence parameters, DKWT achieves an asymptotically nearly optimal worst-case sample complexity.

While the standard DKWT identifies a single generalized Condorcet winner for a single preference relation, our setting involves $D$ distinct users, each with a potentially unique Condorcet winner. Consequently, our modification, detailed in Algorithm 4, executes $D$ sequential tournament instances concurrently. Crucially, to optimize sample efficiency during exploration, our approach leverages shared interactions: a single duel between arms is used to update the empirical margins across all $D$ users simultaneously. This optimization allows the algorithm to concurrently eliminate suboptimal arms for any user as soon as their individual required confidence is satisfied at each round.

In practice, we solve the NSW maximization via the log-transform $\hat{\pi} \in \arg\max_{\pi \in \Delta^K} \sum_{d=1}^{D} \log(\max(\hat{u}_d(\pi), \zeta))$, where $\zeta > 0$ is a small constant added for numerical stability, using Frank-Wolfe run to suboptimality tolerance $\epsilon_{\text{opt}}$ (rather than an exact $\arg\max$).

We establish an upper bound on the total regret of Algorithm 1 in Theorem 4.2.

**Theorem 4.2.** *Consider the fair multi-user dueling bandit problem with $D$ users, $K$ arms, and a time horizon $T$. Suppose user preferences satisfy Assumption 3.1 and the scoring function is defined as in Equation 2. Let $\Delta = \min_{d \in [D], i \neq j} |0.5 - \mathcal{P}_{d,i,j}|$ be the minimum preference gap. If the Fair-ETC algorithm is executed with confidence parameter $\delta = \frac{K \log(K/2)}{2\hat{\Delta}T}$ and exploration length $L = \Theta(K^{-2/3}|\mathcal{A}^*|^{-2/3}D^{\frac{2}{3}}T^{2/3}\log^{1/3}(DKT))$, then for any $\hat{\Delta} \in (0,1)$ such that $\hat{\Delta} > \frac{K \log(K/2)}{2T}$, the expected regret (defined in Equation 4) is upper bounded by $\mathbb{E}[R_T] \leq$:*

$$O\left(\frac{KD}{\Delta^2} \log\left(\frac{K}{2}\right) \left[\log\log\left(\frac{1}{\Delta}\right) + \log\left(\frac{2D\hat{\Delta}T}{K\log(\frac{K}{2})}\right)\right] + D^{\frac{2}{3}}K^{\frac{1}{3}}|\mathcal{A}^*|^{\frac{1}{3}}T^{\frac{2}{3}}\log^{\frac{1}{3}}(DKT) + \frac{K\log(\frac{K}{2})}{\hat{\Delta}}\right).$$

In the context of Theorem 4.2, it is crucial to distinguish between the environment-dependent parameter $\Delta$ and the algorithmic parameter $\hat{\Delta}$. The term $\Delta$ represents the true minimum preference gap inherent to the problem instance; it quantifies the smallest margin by which any user's preference probability deviates from pure chance (0.5). A smaller $\Delta$ indicates a statistically harder problem where preferences are difficult to distinguish. Conversely, $\hat{\Delta}$ is a user-defined hyperparameter in $(0,1)$ used to configure the confidence level $\delta$ of the algorithm. Unlike $\Delta$, which is unknown to the agent, $\hat{\Delta}$ acts as a tuning knob that controls the conservatism of the Condorcet winner identification phase. The theorem holds for any valid choice of $\hat{\Delta}$, but this distinction clarifies that the algorithm does not require knowledge of the true gap $\Delta$ to operate, although the theoretical regret bound is naturally influenced by the interplay between the chosen $\hat{\Delta}$ and the true hardness $\Delta$.

### 4.3 Fair-$\epsilon$-Greedy Algorithm

We design the Fair-$\epsilon$-greedy algorithm for fair multi-user dueling bandits and present it in Algorithm 2. Similar to the Fair-ETC strategy, this algorithm begins with a Condorcet winner identification phase using the Dvoretzky-Kiefer-Wolfowitz tournament algorithm (Haddenhorst et al., 2021). However, instead of separating estimation and optimization into distinct sequential blocks, it balances them in each round. At each time step $t$, with probability $\epsilon_t$, the algorithm explores by playing the next pair of a round-robin schedule over pairs involving the estimated Condorcet winners to refine score estimates. With probability

---

**Algorithm 2** Fair-$\epsilon$-Greedy

---

**Require:** Horizon $T$, Set of Arms $[K]$, Set of Users $[D]$, exploration parameters $\epsilon_t$, sample access to the joint preference tensor $\mathcal{P}$, $\delta$ confidence for DKWT.

1: $(\hat{A}^*, t) \leftarrow \textbf{DKWT}(K, D, \mathcal{P}, \frac{\delta}{D})$.
2: **for** $a^* \in \hat{A}^*$, $a \in \{a \in [K] : a \neq a^*\}$ **do**
3:     Play duel $(a, a^*)$ once and update the estimated $\hat{\mathcal{P}}$.
4:     $t \leftarrow t + 1$.
5: **end for**
6: **for** $t = t + 1$ to $T$ **do**
7:     Sample $u$ uniformly in $[0, 1]$
8:     **if** $u \leq \epsilon_t$ **then**
9:         Play the next pair of a round robin schedule over pairs $a^* \in \hat{A}^*$ and $a \in \{a \in [K] : a \neq a^*\}$ and update the estimated $\hat{\mathcal{P}}$
10:     **else**
11:         Set $\hat{\pi}_t \in \arg\max_{\pi \in \Delta^K} \prod_{d=1}^{D} \mathbb{E}_{a \sim \pi}[2\hat{\mathcal{P}}_d(a, a_d^*)]$.
12:         Sample arms $a_t, a_t'$ independently from $\hat{\pi}_t$.
13:         Observe duels $a_t, a_t'$.
14:     **end if**
15: **end for**

---

$1 - \epsilon_t$, it exploits by playing the policy that maximizes the empirical NSW. We establish an upper bound on the total regret of Algorithm 2 in Theorem 4.3.

Although both algorithms achieve the same asymptotic upper bound, we highlight the following differences. While ETC is simpler and less computationally intensive in implementation, as it requires a single computation of the optimal policy, it cannot recover from poor estimation once the exploitation phase starts. On the other hand, as $\epsilon$-greedy's exploration is carried out across the horizon with a decaying probability, it can still recover from poor early estimates. However, it requires more computation as the current estimated optimal policy is recomputed with each exploitation step. In summary, Fair-ETC is preferred when the environment is stationary, and computation is constrained, while Fair-$\epsilon$-Greedy is preferred when early estimates are unreliable, at the cost of $O(T)$ additional solves.

**Theorem 4.3.** *Consider the same setting as Theorem 4.2. Let $T_0$ denote the number of time steps used for the Condorcet winner identification phase and $\tau_0 = (K - 1)|\mathcal{A}^*|$. If the Fair-$\epsilon$-Greedy algorithm is executed with confidence $\delta = \frac{K \log(K/2)}{2\hat{\Delta}T}$ and a time-decaying exploration rate $\epsilon_t = \Theta\left(D^{2/3}K^{1/3}|\mathcal{A}^*|^{1/3}(t - T_0 - \tau_0)^{-1/3}\log^{1/3}(DK(t - T_0 - \tau_0))\right)$, then for any $\hat{\Delta} \in (0, 1)$ such that $\hat{\Delta} > \frac{K \log(K/2)}{2T}$, the expected regret is upper bounded by $\mathbb{E}[R_T] \leq$:*

$$O\left(\frac{KD}{\Delta^2}\log\left(\frac{K}{2}\right)\left[\log\log\left(\frac{1}{\Delta}\right) + \log\left(\frac{2D\hat{\Delta}T}{K\log(\frac{K}{2})}\right)\right] + D^{\frac{2}{3}}K^{\frac{1}{3}}|\mathcal{A}^*|^{\frac{1}{3}}T^{\frac{2}{3}}\log^{\frac{1}{3}}(DKT) + \frac{K\log(\frac{K}{2})}{\hat{\Delta}}\right).$$

**Regret Bound Gap.** Our upper and lower bounds match in their dependence on $T$ up to logarithmic factors, but not in their dependence on $K$, $D$, and $\Delta$. This gap arises because our upper bounds additively combine a Condorcet winner *identification* cost ($O(KD/\Delta^2)$, inherited from the DKWT subroutine of Haddenhorst et al. (2021)) with an *exploitation* cost for the NSW objective ($O(D^{2/3}K^{1/3}|A^*|^{1/3}T^{2/3})$), while our lower bound isolates only the latter source of hardness, assuming winners are effectively known and depending only on $\min(K, D)$ with no dependence on $\Delta$. The $1/\Delta^2$ term reflects a general winner-identification cost rather than a fairness-specific one, and treating identification and exploitation as separable phases is the reason for this looseness. Closing this gap likely requires an algorithm and a matching lower bound that couple identification and exploitation rather than resolving winners before any welfare-directed exploration begins; we leave this as an open direction for future work.

Table 1: Comparison of final policy performance metrics ($K = 10, D = 10, \Delta = 0.1$).

| Metric | Fair-ETC (Ours) | Fair-$\epsilon$-Greedy (Ours) | Utilitarian | Uniform-Over-Users |
|---|---|---|---|---|
| Cumulative Regret | **201.7 ± 65.8** | 215.7 ± 72.2 | 404.9 ± 85.5 | 577 ± 228.3 |
| Min Welfare | 70545.8 ± 3615.3 | 71798.1 ± 3384.1 | 47101.4 ± 5886.1 | **73567.5 ± 3255.9** |
| Nash Social Welfare | **107370.3 ± 2379** | 106950.1 ± 2359.3 | 103755.9 ± 3001.9 | 98667.9 ± 2113 |
| Gini Coefficient | 0.1176 ± 0.0103 | 0.1164 ± 0.0098 | 0.1998 ± 0.0146 | **0.0883 ± 0.0084** |
| Utilitarian Welfare | 110178.7 ± 2579.4 | 109632.8 ± 2505 | **113472.5 ± 2311.3** | 100076 ± 2066.4 |

**Other Welfare Functions.** We adopt NSW for its axiomatic uniqueness (Kaneko & Nakamura, 1979) and log-concavity. We conjecture our lower bound method could be extended to any welfare function sensitive to a zeroed-out coordinate (e.g., GGF, $p$-means with $p \leq 1$), since the hard instance exploits an informational obstruction rather than NSW's product structure specifically. Our upper bound, however, relies on an NSW-specific sensitivity bound (Lemma A.1) and would require different tools for finding the upper bound for other objectives.

## 5 Experiments

In this section, we validate our theoretical findings through numerical simulations. We compare the performance of our proposed algorithms, Fair-ETC and Fair-$\epsilon$-Greedy, against utilitarian baselines and a balanced uniform strategy. Our experiments aim to answer the following questions: (1) Do our algorithms successfully minimize regret with respect to the NSW objective? (2) How do our algorithms compare to standard utilitarian approaches in terms of fairness metrics such as the Gini coefficient and minimum user utility? (3) How robust are these algorithms when user preferences are clustered, creating distinct majority and minority groups?

### 5.1 Experimental Setup

**Environment Generation.** We simulate a dueling bandit environment with $D$ users and $K$ arms. For each user $d \in [D]$, we generate a preference matrix $\mathcal{P}_d$ satisfying the Condorcet winner assumption. Specifically, for each user, a Condorcet winner $a_d^*$ is either chosen uniformly at random or selected based on the experiment criteria. The pairwise probability $\mathcal{P}_d(a_d^*, j)$ is sampled uniformly from $[0.5 + \Delta, 1.0]$ for all $j \neq a_d^*$, ensuring a minimum gap $\Delta$ to distinguish the winner. All other pairwise probabilities $\mathcal{P}_d(i, j)$ are sampled uniformly from $[0, 1]$, with adjustments to strictly enforce the minimum gap condition. In all experiments, we report the mean and 95% confidence regions over 30 independent runs.

**Algorithms.** We evaluate the following algorithms: (1) **Fair-ETC (Ours):** the Explore-Then-Commit strategy described in Algorithm 1, which optimizes the NSW; (2) **Fair-$\epsilon$-Greedy (Ours):** the $\epsilon$-greedy exploration strategy described in Algorithm 2, which also optimizes NSW; (3) **Utilitarian-ETC & Utilitarian-$\epsilon$-Greedy (Baselines):** variants of the above algorithms that, instead of maximizing the product of utilities (NSW), maximize the *sum* of utilities (Utilitarian Welfare): $f(\pi, s) = \sum_{d=1}^{D} \mathbb{E}[s_d(\pi)]$. These represent standard bandit approaches that ignore fairness; (4) and **Uniform-Over-Users:** a baseline that identifies the Condorcet winner for each user, then at each time step uniformly chooses a user, then plays their winner. Each winner $w$ is played with probability given by $\frac{1}{D} \sum_{d=1}^{D} \mathbb{I}(a_d^* = w)$. Additionally, we use the Frank-Wolfe algorithm to solve the log-concave NSW objective.

In our implementation, we introduce constant scaling factors to the hyperparameters derived in our theoretical analysis to optimize empirical performance. Specifically, we set the exploration length $L$ in Fair-ETC to 0.25 times the theoretical bound. For Fair-$\epsilon$-Greedy, the exploration probability $\epsilon_t$ is scaled by a factor of 0.1. Additionally, the confidence parameter $\hat{\Delta}$ used in the Condorcet winner identification phase is set to 0.0025.

Scaling theoretically derived hyperparameters to improve empirical performance is a standard and well-justified practice in the bandit literature. Theoretical bounds are often conservatively large to account for

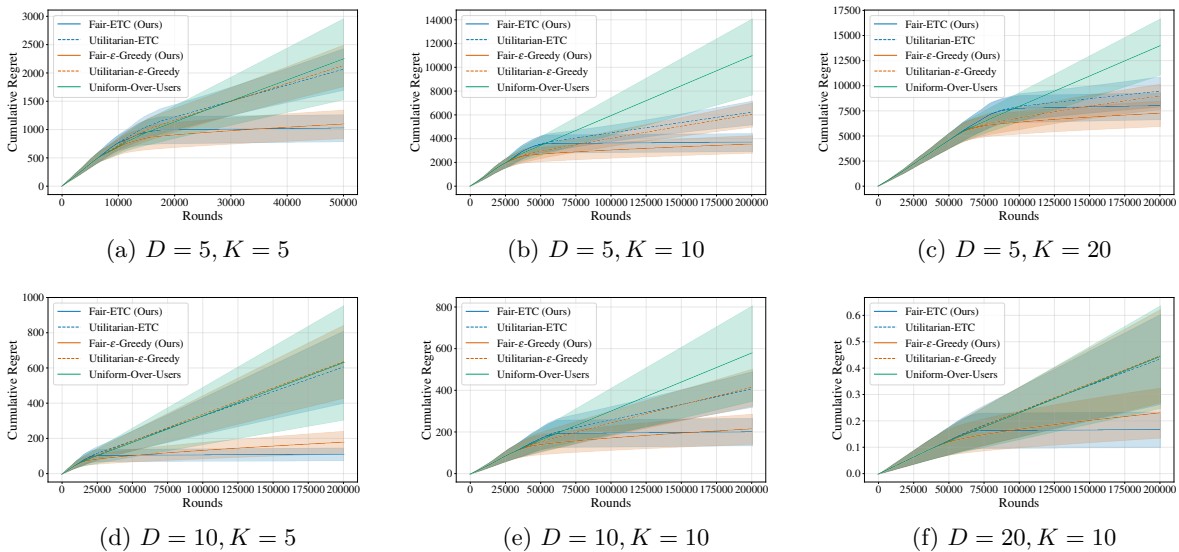

Figure 1: Cumulative NSW Regret across 6 different problem settings varying in Users ($D$), Arms ($K$), and Horizon ($T$). The minimum preference gap is fixed at $\Delta = 0.1$.

worst-case scenarios, making empirical tuning necessary to achieve optimal payoffs in practice. This approach aligns with established methodologies across the field, such as tuning the exploration parameter in LinUCB (Li et al., 2010), calibrating exploration rates for $\epsilon$-greedy and Boltzmann strategies (Kuleshov & Precup, 2014), and applying grid search for algorithm parameters in both neural and contextual bandits (Zhou et al., 2020; Bietti et al., 2021). Thus, introducing these constant scaling factors bridges the gap between our worst-case theoretical guarantees and practical empirical efficiency.

**Metrics.** We compute the utility of each user as the cumulative scores achieved up to round $t$, i.e., $\hat{u}_d(t) = \sum_{\tau=1}^{t} 0.5 \times (s_d(i_\tau) + s_d(j_\tau))$. We assess performance using four key metrics: (1) **Cumulative Regret:** the cumulative difference between the optimal NSW and the NSW of the played policy, as defined in Eq. 4; (2) **Nash Social Welfare:** the geometric mean of user utilities at time T, which is defined as $\prod_{d=1}^{D} \hat{u}_d(T)^{\frac{1}{D}}$; (3) **Minimum User Welfare:** the utility of the worst-off user ($\min_d \hat{u}_d(T)$). Higher values indicate better protection for minority preferences; (4) and **Gini Coefficient:** a measure of inequality among user utilities where 0 represents perfect equality and 1 represents maximal inequality. The Gini coefficient is defined as $G = \frac{\sum_{i=1}^{D} \sum_{j=1}^{D} |\hat{u}_i(T) - \hat{u}_j(T)|}{2D \sum_{k=1}^{D} \hat{u}_k(T)}$.

Note that the NSW appearing in our regret analysis is the product $\prod_d u_d(\pi)$ of expected scores under a single fixed policy $\pi$ and is the one used to compute the regret, whereas the Nash Social Welfare reported here is an additional metric computed from the cumulative realized scores $\hat{u}_d(T)$ over the full trajectory of played policies, used to highlight the fairness of the played trajectory rather than to instantiate directly; we report its $D$-th root (geometric mean) purely for an interpretable, $D$-independent scale, which does not change the ordering over policies.

We solve $\hat{\pi} \in \arg\max_{\pi \in \Delta^K} \sum_{d=1}^{D} \log \max(\hat{u}_d(\pi), \zeta)$ via Frank-Wolfe for $\zeta = 10^{-\frac{1}{12}}$, initialized at the uniform policy, using the standard diminishing step size $\gamma_t = \frac{2}{t+2}$ and the linear minimization oracle over the simplex. We run for a maximum of $T_{\text{FW}} = 300$ iterations, stopping early once the objective value changes by less than $\epsilon_{\text{opt}} = 2 \times 10^{-4}$ between consecutive iterations.

## 5.2 Results: Randomly Generated Preferences

In this section, we investigate the performance of our algorithms in environments where user preferences are generated uniformly at random (subject to the Condorcet constraints). We focus on evaluating the

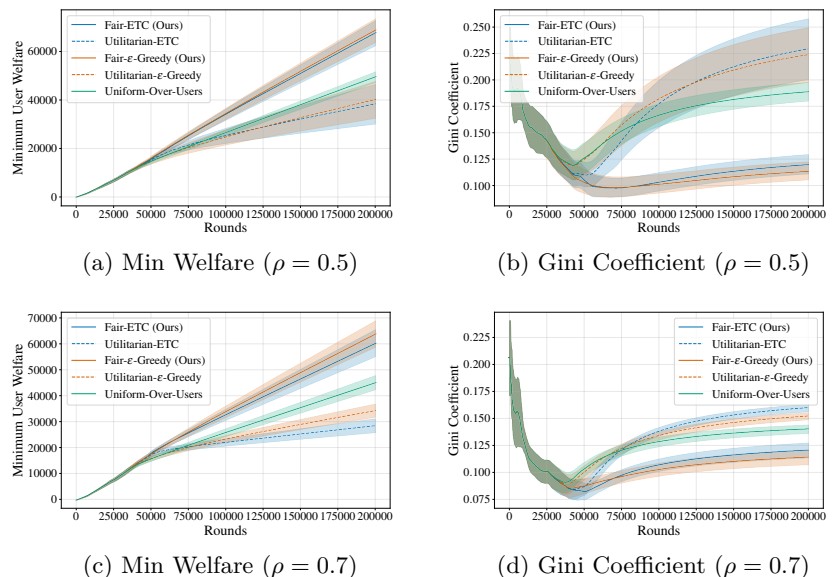

(a) Min Welfare ($\rho = 0.5$)  (b) Gini Coefficient ($\rho = 0.5$)

(c) Min Welfare ($\rho = 0.7$)  (d) Gini Coefficient ($\rho = 0.7$)

Figure 2: Fairness metrics under clustered user preferences ($D = 10, K = 10$). We vary the clustering fraction $\rho$ (majority size). The utilitarian baseline exhibits high inequality (high Gini, low Min Welfare) as the majority cluster grows, whereas fair algorithms remain robust.

cumulative NSW regret across a diverse set of problem configurations defined by the tuple $(D, K, T)$, fixing the minimum preference gap at $\Delta = 0.1$.

Figure 1 displays the cumulative regret for six distinct scenarios ranging from small-scale settings ($D = 5, K = 5$) to larger environments ($D = 20, K = 10$). Across all settings, **Fair-ETC** and **Fair-$\epsilon$-Greedy** demonstrate sublinear regret, consistently outperforming the utilitarian and uniform-over-users baselines.

We further analyze the quality of the learned policies in Table 1, which compares the algorithms across four distinct metrics: minimum user welfare, Nash social welfare, Gini coefficient, and utilitarian welfare. We observe a clear trade-off between aggregate efficiency and fairness. For each metric, we report only one utilitarian algorithm, which is the best performing for this metric.

The Utilitarian algorithms achieve high utilitarian welfare, as expected, but this comes at the cost of significantly higher inequality, evidenced by high Gini coefficients, lower minimum user welfare, and lower NSW. In contrast, our proposed algorithms, **Fair-ETC** and **Fair-$\epsilon$-Greedy**, achieve the highest Nash Social Welfare. In terms of the other fairness metrics, specifically minimum user welfare and the Gini coefficient, our algorithms perform significantly better than the utilitarian approaches and remain close to the Uniform-Over-Users policy, which is a naturally balanced (though inefficient) baseline. This demonstrates that the Nash Social Welfare objective successfully optimizes for efficiency while maintaining fairness levels comparable to an unbiased random strategy.

## 5.3   Results: Clustered Preferences

In this section, we examine the robustness of our algorithms in a polarized setting where users are divided into majority and minority clusters. We simulate a scenario with $D = 10$ users and $K = 10$ arms over a horizon of $T = 200,000$. A fraction of users (controlled by the parameter $\rho \in \{0.5, 0.7\}$) share a single Condorcet winner $w$, forming a majority, while the remaining users have randomly assigned preferences with $s_d(w) < 1$ for users outside of the majority.

Figure 2 illustrates the impact of this clustering on fairness metrics. The top and bottom rows display the fairness metrics for $\rho = 0.5$ and $\rho = 0.7$ respectively. The utilitarian algorithms increasingly favor the majority's preference at the expense of the minority, leading to lower minimum welfare and higher inequality. In

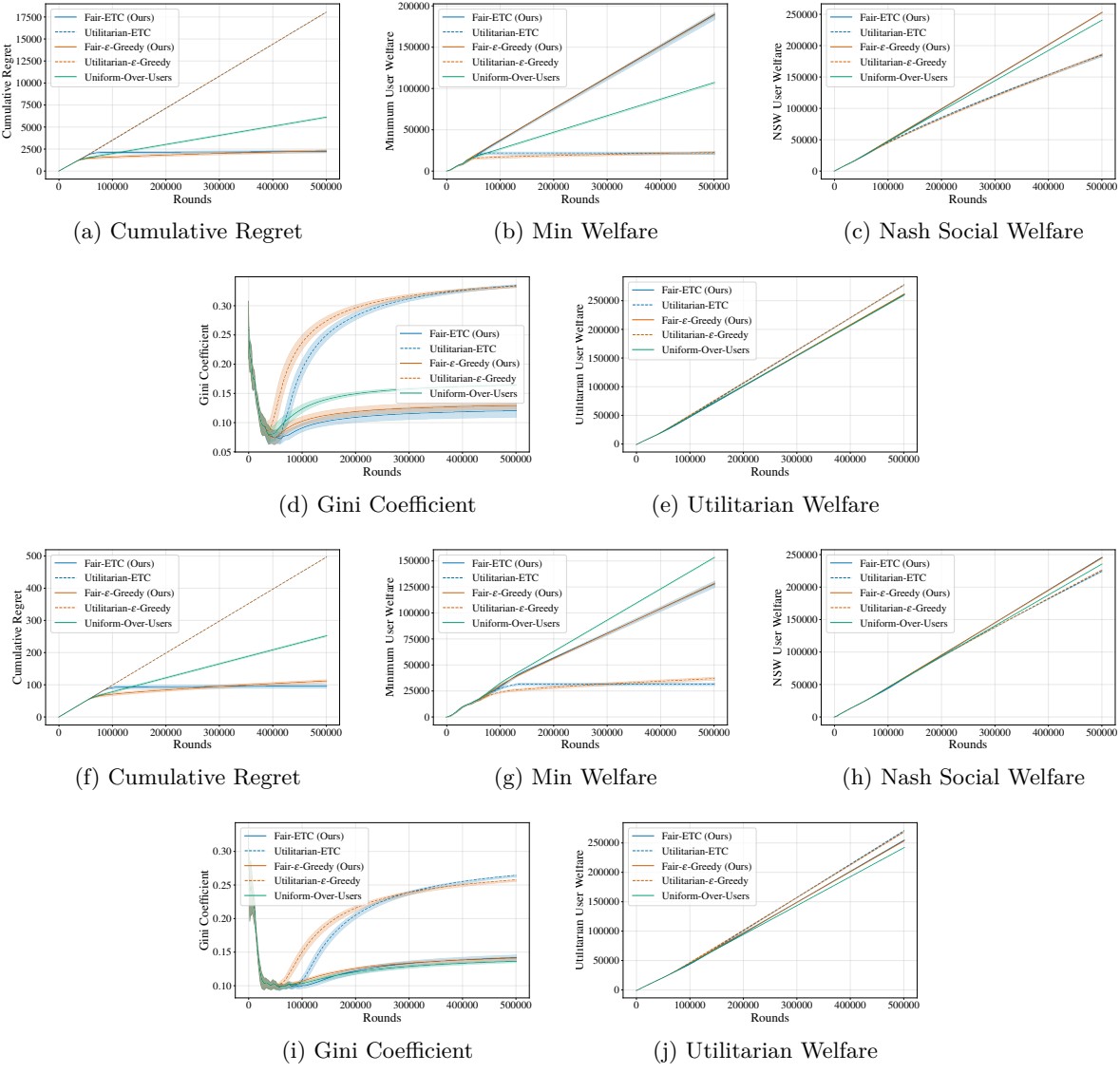

Figure 3: Performance metrics on the Sushi dataset configured with $K = 10$ arms. Top two rows: $D = 5$ clusters. Bottom two rows: $D = 10$ clusters. Our proposed methods achieve the lowest cumulative regret in both settings, while maintaining highly competitive fairness metrics.

contrast, Fair-ETC and Fair-$\epsilon$-Greedy maintain high minimum welfare and low Gini coefficients, successfully balancing the interests of users and outperforming the Uniform-Over-Users policy.

## 5.4 Results: Sushi Dataset

To evaluate our algorithms on real-world human preferences, we utilize the Sushi dataset (Kamishima, 2003), which contains complete rankings of $K = 10$ sushi variants from 5,000 individuals. To map these individual rankings into our multi-user dueling bandit framework, we group the individuals into $D$ macroscopic users (clusters) representing distinct preference profiles.

**Data Preprocessing.** To measure the similarity between individual tastes, we compute the Spearman correlation $r$ (Marden, 1996) between all pairs of valid rankings and define the pairwise distance as $(1 - r)/2$. Using this precomputed distance matrix, we apply Agglomerative Clustering with average linkage to partition

the dataset into $D \in \{5, 10\}$ clusters (Hastie, 2009). For each cluster $d \in [D]$, the pairwise preference probability $\mathcal{P}_{d,i,j}$ is calculated as the fraction of individuals within that cluster who ranked item $i$ strictly higher than item $j$.

To ensure the resulting problem instance satisfies our unique Condorcet winner assumption (Assumption 3.1) with a statistically distinguishable margin, we enforce a minimum preference gap of $\Delta = 0.1$. Any pairwise probability $\mathcal{P}_{d,i,j}$ falling within the $(0.4, 0.6)$ range is projected to the nearest boundary of $\{0.4, 0.6\}$, while preserving the reciprocal property $\mathcal{P}_{d,i,j} + \mathcal{P}_{d,j,i} = 1$. We confirm that this preprocessing step yields a unique Condorcet winner in every cluster, in both the $D = 5$ and $D = 10$ configurations, with the enforced minimum preference gap of $\Delta = 0.1$ satisfying Assumption 3.1 throughout, and that no cluster required further adjustment beyond the boundary projection described above.

**Results.** We evaluate the algorithms over the $D = 5$ and $D = 10$ cluster configurations, presenting the cumulative regret, Nash Social Welfare, minimum user welfare, Gini coefficient, and utilitarian welfare in Figure 3. Across both configurations, our proposed methods (Fair-ETC and Fair-$\epsilon$-Greedy) achieve the lowest cumulative regret. In the $D = 5$ setting, the proposed methods strongly dominate across all fairness metrics, achieving the lowest Gini coefficient and highest minimum user welfare. As the environment becomes more fragmented in the $D = 10$ setting, the uniform-over-users baseline achieves slightly better fairness metrics (Min Welfare and Gini coefficient); however, this marginal gain in fairness comes at the cost of vastly inferior cumulative regret. Overall, the proposed algorithms consistently maintain a better balance, effectively minimizing regret while protecting minority welfare.

## 6 Conclusion

In this work, we presented the study of fairness in dueling bandits with heterogeneous user preferences. By adopting the NSW as our objective, we provided a framework for balancing competing interests without marginalizing minority groups. We established a fundamental regret lower bound of $\Omega(T^{2/3} \min(K, D)^{\frac{1}{3}})$ for this setting and developed two algorithms, Fair-ETC and Fair-$\epsilon$-Greedy. Our extensive empirical evaluation confirms that these fairness-aware algorithms significantly reduce inequality compared to standard utilitarian approaches, as evident by the Gini coefficient and minimum welfare metrics, offering protection to users with distinct preferences while maintaining efficient learning.

Future research avenues include extending this framework to contextual dueling bandits and investigating the impact of continuous action spaces on fairness guarantees. Furthermore, we aim to consider settings where the Condorcet winners assumption does not hold. In such settings, one can potentially resort to utilizing metrics such as Copeland (Zoghi et al., 2015) and Borda scores (Jamieson et al., 2015). Finally, another promising direction is to explore scenarios with sparse feedback, where each played pair of arms is evaluated by only a subset of users.

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

## A  Missing Proofs

This appendix presents the detailed proofs for the theoretical results established in the main paper.

### A.1  Proofs for Section 4.1

**Proof of Theorem 4.1**

*Proof.* We construct a set of problem instances and show that no algorithm can yield lower regret. We start by designing instance $\mathcal{I}^0$ and constructing $\frac{\min(K,D)}{2}$ perturbations and analyze the average regret across the $\frac{\min(K,D)}{2}$ instances.

Assume $K, D \geq 4$ and even (the construction can be adapted to odd values).

**Condorcet winners.**  Let $\mathcal{A}^*$ be the set of Condorcet winners. For each user $d \in \{1, \dots, D\}$ the Condorcet winner is

$$a_d^* = a_{(d \bmod K)+1}.$$

**Partition of winners.**  Let $K^* := |\mathcal{A}^*|$ and choose a subset $\bar{\mathcal{A}} \subset \mathcal{A}^*$ with $|\bar{\mathcal{A}}| = K^*/2$. Write $\bar{\mathcal{A}}$ for the set of *good winners* and $\mathcal{A}^* \setminus \bar{\mathcal{A}}$ for the *bad winners*. A bad winner is a Condorcet winner for one user but loses with probability 1 to Condorcet winners of other users. Hence, playing this arm with probability $p$ yields a utility of $U_d(\pi) \leq (1-p)$ for other users. In addition, let $\bar{D} = \{d \in [D] : a_d^* \in \bar{\mathcal{A}}\}$.

**Instances.**  We will define $|\bar{\mathcal{A}}|$ problem instances; index them by $m \in \{1, \dots, |\bar{\mathcal{A}}|\}$ and denote the $m$-th instance by $\mathcal{I}^m$. We denote the arm $a_m$ as the $m$-th good winner of the set $\bar{\mathcal{A}}$ and is the one of interest in instance $\mathcal{I}^m$.

**Probability relationships (for each user $d$).**  Fix small parameters $\epsilon \in (0, 0.2)$ and $\epsilon' \in (0, \min(\frac{1}{2T^{\alpha+1}D}, 0.05))$ for some $\alpha \geq 2$. $\epsilon$ can be viewed as the perturbation between instance $\mathcal{I}^0$ and instances $\mathcal{I}^m$. However, $\epsilon'$ is an arbitrarily small positive number that separates a Condorcet winner $a_d^*$ from other close arms such that $\mathcal{P}_d(a_d^*, a) > 0.5$ and does not affect the bound.

We list the preference relation between arms below:

- $\bar{\mathcal{A}}$ **vs.** $\mathcal{A}^* \setminus \bar{\mathcal{A}}$: For any $a_d^* \in \bar{\mathcal{A}}$ and $a_{d'}^* \in \mathcal{A}^* \setminus \bar{\mathcal{A}}$:

$$\mathcal{P}_d(a_d^*, a_{d'}^*) = 1$$

- $\bar{\mathcal{A}}$ **vs.** $\bar{\mathcal{A}}$: For any $a_d^*, a \in \bar{\mathcal{A}}$ with $a \neq a_d^*$:

$$\mathcal{P}_d(a_d^*, a) = 1/2 + \epsilon'$$

- $\mathcal{A}^* \setminus \bar{\mathcal{A}}$ **vs.** $\mathcal{A}^* \setminus \bar{\mathcal{A}}$: For any $a_d^*, a \in \mathcal{A}^* \setminus \bar{\mathcal{A}}$ with $a_d^* \neq a$:

$$\mathcal{P}_d(a_d^*, a) = 1$$

- $\mathcal{A}^*$ **vs. Non-$\mathcal{A}^*$:** For any $a_d^* \in \mathcal{A}^*$ with $a \notin \mathcal{A}^*$:

$$\mathcal{P}_d(a_d^*, a) = 1$$

- $\mathcal{A}^* \setminus \bar{\mathcal{A}}$ **vs.** $\bar{\mathcal{A}}$: For any $a_d^* \in \mathcal{A}^* \setminus \bar{\mathcal{A}}$ and $a_{d'}^* \in \bar{\mathcal{A}}$:

$$\mathcal{P}_d(a_d^*, a_{d'}^*) = \begin{cases} 1/2 + \epsilon' + \epsilon, & d' \neq m \\ 1/2 + \epsilon', & d' = m \end{cases}$$

Table 2: Scores $s_d$ under instance $\mathcal{I}_m$

| | $s_{d'}$ s.t. $d' \in \bar{D}$ | $s_{d'}$ s.t. $d' \in [D] \setminus \bar{D}$ |
|---|---|---|
| $a_d^* \in \bar{\mathcal{A}}$ | $\begin{cases} 1, & d = d' \\ 1 - 2\epsilon', & d \neq d' \end{cases}$ | $\begin{cases} 1 - 2\epsilon', & a_d^* = a_m \\ 1 - 2(\epsilon' + \epsilon), & a_d^* \neq a_m \end{cases}$ |
| $a_d^* \in \mathcal{A}^* \setminus \bar{\mathcal{A}}$ | $0$ | $\begin{cases} 1, & d = d' \\ 0, & d \neq d' \end{cases}$ |
| $a \notin \mathcal{A}^*$ | $0$ | $0$ |

Note that the preference values for the reverse pairs are given by $\mathcal{P}_d(j,i) = 1 - \mathcal{P}_d(i,j)$. Recall the definition of the score being $s_d(i) = 2\mathcal{P}_d(i, a_d^*)$. The scores of the arms are summarized in Table 2.

For instance $\mathcal{I}^0$, any policy $\pi$ with support $\bar{\mathcal{A}}$ is optimal with $NSW(\pi, s) = (1 - 2\epsilon')^{\frac{D}{2}-1}(1 - 2\epsilon' - 2\epsilon)^{\frac{D}{2}}$. However, it is easy to show that for instance $\mathcal{I}^m$, only a deterministic policy is optimal with $\pi(a_m) = 1$ and $NSW(\pi, s) = (1 - 2\epsilon')^{D-1}$. However, in order to distinguish the arm $m$, good winners in $\bar{\mathcal{A}}$ need to be compared with bad winners in $\mathcal{A}^* \setminus \bar{\mathcal{A}}$ to identify the best good winner $m$.

Before beginning with the proof, we note two important remarks.

**Remark 1:** For any policy $\pi$ with $\pi(a_d^*) = \delta$, $\delta > 0$ and $a_d^* \in \mathcal{A}^* \setminus \bar{\mathcal{A}}$, create $\pi'$ with $\pi'(a_d^*) = 0$ and $\pi'(a_{d'}^*) = \pi(a_{d'}^*) + \delta$ for some $d' \in \bar{D}$ and $a_{d'}^* \neq a_m$. Policy $\pi'$ satisfies

$$NSW(\pi', s) \geq NSW(\pi, s).$$

To show that, we consider the following. For $d''$ with $a_{d''}^* \neq a_d^*$, $s_{d''}(a_{d'}^*) \geq s_{d''}(a_d^*)$. Therefore, $\mathbb{E}_{a \sim \pi'}[s_{d''}(a)] \geq \mathbb{E}_{a \sim \pi}[s_{d''}(a)] \, \forall d''$ with $a_{d''}^* \neq a_d^* \neq a_{d'}^*$. Hence, in order to show that, $NSW(\pi', s) \geq NSW(\pi, s)$, it is sufficient to show that

$$\left(\mathbb{E}_{a \sim \pi'}[s_i(a)]\right)\left(\mathbb{E}_{a \sim \pi'}[s_j(a)]\right) \geq \left(\mathbb{E}_{a \sim \pi}[s_i(a)]\right)\left(\mathbb{E}_{a \sim \pi}[s_j(a)]\right), \tag{5}$$

for unique pairs, $\{i, j \in [D] : a_i^* = a_{d'}^* \text{ and } a_j^* = a_d^*\}$. Due to the way the matrices are set, the cardinality of the set $\{i : a_i^* = a_{d'}^*\}$ is greater than or equal to the cardinality of the set $\{j : a_j^* = a_d^*\}$, and we can construct such pairs.

Let $C_{d'} = \sum_{a \in [K], a \neq a_d^*} s_{d'}(a)\pi(a)$. Noting that $-2(\epsilon + \epsilon') \geq -0.5$, we can write the left hand side of Equation 5 as

$$\left(\mathbb{E}_{a \sim \pi'}[s_i(a)]\right)\left(\mathbb{E}_{a \sim \pi'}[s_j(a)]\right) \geq (C_i + \delta)(C_j + \delta(1 - 0.5))$$

$$= C_i(C_j + \delta) + \delta(C_j - 0.5C_i) + 0.5\delta^2$$
$$= \left(\mathbb{E}_{a \sim \pi}[s_i(a)]\right)\left(\mathbb{E}_{a \sim \pi}[s_j(a)]\right) + \delta(C_j - 0.5C_i) + 0.5\delta^2$$
$$\geq \left(\mathbb{E}_{a \sim \pi}[s_i(a)]\right)\left(\mathbb{E}_{a \sim \pi}[s_j(a)]\right),$$

where the last inequality follows from the fact that $s_j(a) - 0.5s_i(a) \geq 0 \, \forall a \in [K]$. This remark allows us to use the NSW of policies that only play arms in $\bar{\mathcal{A}}$ as an upper bound to the NSW of other policies.

**Remark 2:** If an algorithm chooses policies with either of the following conditions $\sum_{t=1}^{T} \sum_{a_d^* \in \mathcal{A}^* \setminus \bar{\mathcal{A}}} Pr_{\mathcal{I}^0}(i_t = a_d^*) \geq \epsilon T$ or $\sum_{t=1}^{T} \sum_{a_d^* \in \mathcal{A}^* \setminus \bar{\mathcal{A}}} Pr_{\mathcal{I}^0}(j_t = a_d^*) \geq \epsilon T$, where $(i_t, j_t)$ is the pair played at time $t$, then the algorithm will incur regret $\Omega(\epsilon T)$.

Since arms $a_d^* \in \mathcal{A}^* \setminus \bar{\mathcal{A}}$, have zero scores for users $d' : a_{d'}^* \neq a_d^*$, we can upper bound the utility of user $d' \in \bar{D}$ for any policy $\pi^t$ as follows,

$$\mathbb{E}_{a \sim \pi^t}[s_{d'}(a)] \leq \left(1 - \sum_{a_d^* \in \mathcal{A}^* \setminus \bar{\mathcal{A}}} \pi^t(a_d^*)\right),$$

hence $NSW(\pi^t, s) \leq \left(1 - \sum_{a_d^* \in \mathcal{A}^* \setminus \bar{\mathcal{A}}} \pi^t(a_d^*)\right)$

We consider the policy $\pi_m^*$ that plays arm $m$ with probability 1 for instance $\mathcal{I}^m$, with $NSW(\pi_m^*, s) = (1 - 2\epsilon')^{D-1}$.

The instantaneous regret at time $t$ is upper bounded by

$$r_t \geq \frac{1}{2}\left(NSW(\pi_m^*, s) - NSW(\pi^t, s)\right) \geq \frac{1}{2}\left((1 - 2\epsilon')^{D-1} - 1\right) + \frac{1}{2}\sum_{a_d^* \in \mathcal{A}^* \setminus \bar{\mathcal{A}}} \pi^t(a_d^*) \geq -\epsilon' D + \frac{1}{2}\sum_{a_d^* \in \mathcal{A}^* \setminus \bar{\mathcal{A}}} \pi^t(a_d^*),$$

where the last inequality follows from the Bernoulli inequality.

Therefore, the total expected regret can be lower bounded as follows:

$$\mathbb{E}_{\mathcal{I}^0}[R_T] \geq -\epsilon' DT + \frac{1}{2}\sum_{t=1}^{T} \sum_{a_d^* \in \mathcal{A}^* \setminus \bar{\mathcal{A}}} Pr_{\mathcal{I}^0}(i_t = a_d^*) \geq -\frac{1}{2T^\alpha} + \frac{\epsilon T}{2} = \Omega(\epsilon T),$$

as the first term can be made arbitrarily small compared to the second term. This remark allows us to WLOG assume that $\sum_{t=1}^{T} \sum_{a_d^* \in \mathcal{A}^* \setminus \bar{\mathcal{A}}} Pr_{\mathcal{I}^0}(i_t = a_d^*) \leq \epsilon T$ and $\sum_{t=1}^{T} \sum_{a_d^* \in \mathcal{A}^* \setminus \bar{\mathcal{A}}} Pr_{\mathcal{I}^0}(j_t = a_d^*) \leq \epsilon T$ as otherwise the expected regret will be at least $\epsilon T$.

We now analyze the lower bound of the average regret across instances. Again, consider the policy $\pi_m^*$ that plays arm $m$ with probability 1 for instance $\mathcal{I}^m$, with $NSW(\pi_m^*, s) = (1 - 2\epsilon')^{D-1}$.

Due to remark 1, WLOG we can use the Nash social welfare of policies that do not play arms $\in \mathcal{A}^* \setminus \bar{\mathcal{A}}$ as an upper bound to the NSW of policies that play arms $\in \mathcal{A}^* \setminus \bar{\mathcal{A}}$. For any policy $\pi$ such that $\pi(m) \leq 1$ and $\pi(a_d^*) = 0 \; \forall a_d^* \in \mathcal{A}^* \setminus \bar{\mathcal{A}}$, there exists at least one user $d$ such that $a_d^* \in \mathcal{A}^* \setminus \bar{\mathcal{A}}$ for which

$$\mathbb{E}_{a \sim \pi}\left[s_d(a)\right] \leq \pi(m)(1 - 2\epsilon') + (1 - \pi(m))(1 - 2\epsilon' - 2\epsilon) = 1 - 2\epsilon' - 2(1 - \pi(m))\epsilon \leq 1 - 2(1 - \pi(m))\epsilon$$

Therefore, the Nash Social Welfare satisfies

$$NSW(\pi, s) \leq (1 - 2(1 - \pi(m))\epsilon).$$

The instantaneous regret of round $t$ can thus be lower bounded as

$$r_t \geq \left((1 - 2\epsilon')^{D-1} - 1\right) + \epsilon\left(2 - \pi^t(m) - \pi'^t(m)\right) \geq -2\epsilon' D + \epsilon\left(2 - \pi^t(m) - \pi'^t(m)\right),$$

where the second inequality comes from the Bernoulli inequality.

Averaging the expected total regret across all instances, we get,

$$\sum_{m=1}^{|\bar{\mathcal{A}}|} \frac{\mathbb{E}_{\mathcal{I}^m}[R_T]}{|\bar{\mathcal{A}}|} = \sum_{m=1}^{|\bar{\mathcal{A}}|} \sum_{t=1}^{T} \frac{\mathbb{E}_{\mathcal{I}^m}[r_t]}{|\bar{\mathcal{A}}|} \geq -2\epsilon' DT + 2\epsilon T - \epsilon \sum_{m=1}^{|\bar{\mathcal{A}}|} \sum_{t=1}^{T} \frac{\mathbb{E}_{\mathcal{I}^m}[\pi^t(m) + \pi'^t(m)]}{|\bar{\mathcal{A}}|} \tag{6}$$

Now as $|\pi^t(m) + \pi'^t(m)| \leq 2$, it follows that

$$\sum_{m=1}^{|\bar{\mathcal{A}}|} \sum_{t=1}^{T} \left(\frac{\mathbb{E}_{\mathcal{I}^m}[\pi^t(m) + \pi'^t(m)]}{|\bar{\mathcal{A}}|} - \frac{\mathbb{E}_{\mathcal{I}^0}[\pi^t(m) + \pi'^t(m)]}{|\bar{\mathcal{A}}|}\right) \leq \sum_{m=1}^{|\bar{\mathcal{A}}|} \frac{4T D_{TV}(\mathcal{I}^m, \mathcal{I}^0)}{|\bar{\mathcal{A}}|},$$

where $D_{TV}(\mathcal{I}^m, \mathcal{I}^0)$ is the total variation distance between $\mathcal{I}^m$ and $\mathcal{I}^0$ with respect to the sigma algebra $\mathcal{H}_T$ defined over the observed history, i.e. $\mathcal{H}_T = \sigma(\{P_t(i_t, j_t)\}_{t \in T})$ and $D_{TV}(\mathcal{I}^m, \mathcal{I}^0) = \sup_{\mathcal{E} \in \mathcal{H}_t} |Pr_{\mathcal{I}^m}(\mathcal{E}) - Pr_{\mathcal{I}^0}(\mathcal{E})|$.

Using Pinsker's inequality, we get,

$$\sum_{m=1}^{|\bar{\mathcal{A}}|} \sum_{t=1}^{T} \left( \frac{\mathbb{E}_{\mathcal{I}^m}[\pi^t(m) + \pi'^{\,t}(m)]}{|\bar{\mathcal{A}}|} - \frac{\mathbb{E}_{\mathcal{I}^0}[\pi^t(m) + \pi'^{\,t}(m)]}{|\bar{\mathcal{A}}|} \right) \leq \sum_{m=1}^{|\bar{\mathcal{A}}|} \frac{4T D_{TV}(\mathcal{I}^m, \mathcal{I}^0)}{|\bar{\mathcal{A}}|}$$

$$\leq \sum_{m=1}^{|\bar{\mathcal{A}}|} \frac{4T \sqrt{0.5 D_{KL}(\mathcal{I}^0, \mathcal{I}^m)}}{|\bar{\mathcal{A}}|} \qquad (7)$$

$$\leq 4T \sqrt{\frac{\sum_{m=1}^{|\bar{\mathcal{A}}|} D_{KL}(\mathcal{I}^0, \mathcal{I}^m)}{2|\bar{\mathcal{A}}|}},$$

where $D_{KL}$ is the Kullback–Leibler divergence between $\mathcal{I}^0$ and $\mathcal{I}^m$, and the last inequality follows from Jensen's inequality.

Using the chain rule of KL-divergence, we have,

$$\sum_{m=1}^{|\bar{\mathcal{A}}|} D_{KL}(\mathcal{I}^0, \mathcal{I}^m) \leq \sum_{m=1}^{|\bar{\mathcal{A}}|} \sum_{t=1}^{T} D_{KL}(\mathcal{I}_t^0, \mathcal{I}_t^m),$$

where $\mathcal{I}_t^m = Pr_{\mathcal{I}^m}(i_t, j_t | \mathcal{H}_{t-1})$.

Now note that,

$$D_{KL}(\mathcal{I}_t^0, \mathcal{I}_t^m) = \begin{cases} D_{KL}(Ber(0.5 + \epsilon'), Ber(0.5 + \epsilon + \epsilon')), & if\,(i_t, j_t) = (a_d^*, m) \\ D_{KL}(Ber(0.5 - \epsilon'), Ber(0.5 - \epsilon - \epsilon')), & if\,(i_t, j_t) = (m, a_d^*) \\ 0, & otherwise \end{cases}$$

where $a_d^* \in \mathcal{A}^* \setminus \bar{\mathcal{A}}$ and $Ber(p)$ is a Bernoulli random variable with mean $p$. For $\epsilon' < 0.05$,

$$D_{KL}(Ber(0.5 + \epsilon'), Ber(0.5 + \epsilon + \epsilon')) = D_{KL}(Ber(0.5 - \epsilon'), Ber(0.5 - \epsilon - \epsilon')) \leq \frac{\epsilon^2}{0.25 - (\epsilon + \epsilon')^2} \leq 6\epsilon^2.$$

It follows that

$$\sum_{m=1}^{|\bar{\mathcal{A}}|} D_{KL}(\mathcal{I}^0, \mathcal{I}^m) \leq \sum_{m=1}^{|\bar{\mathcal{A}}|} \sum_{t=1}^{T} D_{KL}(\mathcal{I}_t^0, \mathcal{I}_t^m)$$

$$\leq 6\epsilon^2 \sum_{t=1}^{T} \sum_{a_d^* \in \mathcal{A}^* \setminus \bar{\mathcal{A}}} \sum_{m=1}^{|\bar{\mathcal{A}}|} (Pr_{\mathcal{I}^0}(i_t = a_d^*, j_t = m) + Pr_{\mathcal{I}^0}(i_t = m, j_t = a_d^*))$$

$$\leq 6\epsilon^2 \sum_{t=1}^{T} \sum_{a_d^* \in \mathcal{A}^* \setminus \bar{\mathcal{A}}} (Pr_{\mathcal{I}^0}(i_t = a_d^*) + Pr_{\mathcal{I}^0}(j_t = a_d^*)) \leq 12\epsilon^3 T,$$

where the last inequality follows from remark 2.

Plugging it back in Equation 7 we get,

$$\sum_{m=1}^{|\bar{\mathcal{A}}|} \sum_{t=1}^{T} \frac{\mathbb{E}_{\mathcal{I}^m}[\pi^t(m) + \pi'^{\,t}(m)]}{|\bar{\mathcal{A}}|} \leq 4T \sqrt{\frac{12\epsilon^3 T}{2|\bar{\mathcal{A}}|}} + \sum_{m=1}^{|\bar{\mathcal{A}}|} \sum_{t=1}^{T} \frac{\mathbb{E}_{\mathcal{I}^0}[\pi^t(m) + \pi'^{\,t}(m)]}{|\bar{\mathcal{A}}|}$$

$$\leq 4T \sqrt{\frac{12\epsilon^3 T}{2|\bar{\mathcal{A}}|}} + \frac{2T}{|\bar{\mathcal{A}}|}$$

Plugging it back in Equation 6 we get,

$$
\sum_{m=1}^{|\bar{\mathcal{A}}|} \frac{\mathbb{E}_{\mathcal{I}^m}[R_T]}{|\bar{\mathcal{A}}|} = \sum_{m=1}^{|\bar{\mathcal{A}}|} \sum_{t=1}^{T} \frac{\mathbb{E}_{\mathcal{I}^m}[r_t]}{|\bar{\mathcal{A}}|} \geq -2\epsilon' D T + \epsilon \left( 2T - 4T\sqrt{\frac{12\epsilon^3 T}{|\mathcal{A}^*|}} - \frac{4T}{|\mathcal{A}^*|} \right)
$$
$$
\geq -\frac{1}{T^\alpha} + \epsilon \left( 2T - 4T\sqrt{\frac{12\epsilon^3 T}{|\mathcal{A}^*|}} - T \right),
$$

(8)

where the last inequality follows from $\epsilon' \leq \frac{1}{2T^3 D}$ and $|\mathcal{A}^*| \geq 4$. We now consider two cases.

**Case 1:** When $T \leq \frac{|\mathcal{A}^*|}{768\epsilon^3}$. Equation 8 becomes

$$
\sum_{m=1}^{|\bar{\mathcal{A}}|} \frac{\mathbb{E}_{\mathcal{I}^m}[R_T]}{|\bar{\mathcal{A}}|} \geq -\frac{1}{T^\alpha} + \epsilon \frac{T}{2} = \Omega(\epsilon T),
$$

As $-\frac{1}{T^\alpha}$ can be made arbitrarily small compared to the second term. Now as $\epsilon \leq \left( \frac{|\mathcal{A}^*|}{768} \right)^{\frac{1}{3}}$, we can set $\epsilon = \alpha \left( \frac{|\mathcal{A}^*|}{768} \right)^{\frac{1}{3}}$ for some $\alpha \in (0,1)$ and we get

$$
\sum_{m=1}^{|\bar{\mathcal{A}}|} \frac{\mathbb{E}_{\mathcal{I}^m}[R_T]}{|\bar{\mathcal{A}}|} = \Omega(T^{\frac{2}{3}} |\mathcal{A}^*|^{\frac{1}{3}})
$$

**Case 2:** When $T > \frac{|\mathcal{A}^*|}{768\epsilon^3}$. We prove that $\mathbb{E}[R_T] \geq \frac{|\mathcal{A}^*|}{1536\epsilon^2}$ by contradiction. Let $T_0 = \frac{|\mathcal{A}^*|}{768\epsilon^3}$. Assume there exists $T' > T_0$ with $\mathbb{E}[R_T] \leq \frac{|\mathcal{A}^*|}{1536\epsilon^2}$. However, this implies $\mathbb{E}[R_{T_0}] \leq \mathbb{E}[R_{T'}] < \frac{|\mathcal{A}^*|}{1536\epsilon^2} = \frac{\epsilon T_0}{2}$ which is a contradiction by case 1. Therefore, $\mathbb{E}[R_T] = \Omega \left( \frac{|\mathcal{A}^*|}{\epsilon^2} \right)$. Similar to case 1, as $\frac{1}{\epsilon^2} < \left( \frac{768T}{|\mathcal{A}^*|} \right)^{\frac{2}{3}}$, we can set $\epsilon$ such that $\mathbb{E}[R_T] = \Omega(T^{\frac{2}{3}} |\mathcal{A}^*|^{\frac{1}{3}})$. And as $|\mathcal{A}^*| = min(K,D)$ in this example, we get the Theorem statement. $\square$

## A.2 Proof for Section 4.2

We first present a useful Lemma.

**Lemma A.1.** *(Hossain et al., 2021) Given a policy $\pi \in \Delta^K$ score Matrices $s^1, s^2 \in [0,1]^{D \times K}$ , we have*

$$
|NSW(\pi, s^1) - NSW(\pi, s^2)| \leq \sum_{d=1}^{D} \sum_{i=1}^{K} \pi(i) |s^1_{d,i} - s^2_{d,i}|
$$

**Proof of Theorem 4.2**

*Proof.* Let $\mathbb{E}[T_d]$ be the expected number of steps the DKWT algorithm identifies the Condorcet winner for user $d$. The expected regret can then be written as

$$
\mathbb{E}[R_T] \leq \sum_{d=1}^{D} \mathbb{E}[T_d] + KL|\mathcal{A}^*| + \left( T - KL|\mathcal{A}^*| - \sum_{d=1}^{D} \mathbb{E}[T_d] \right) \mathbb{E}[NSW(\pi^*, s) - NSW(\hat{\pi}, s)]
$$

The first term corresponds to the regret accumulated during the identification phase, while the second term accounts for the exploration phase, and the final term is the regret of the commit phase.

Define the events:

1. $\mathcal{E}_1$: The DKWT algorithm correctly identifies the Condorcet winner for all users $d \in [D]$ during Phase 1. Formally, $\mathcal{E}_1 = \{\forall d \in [D] : \hat{a}^*_d = a^*_d\}$.

2. $\mathcal{E}_2$: The estimated score of all arms is sufficiently close to the true score. Formally, $\mathcal{E}_2 = \left\{ \forall d \in [D], a \in [K] : |\hat{s}_d(a) - s_d(a)| \leq \sqrt{\frac{\log(DKT)}{L}} \right\}$.

Let $\mathcal{E} = \mathcal{E}_1 \cap \mathcal{E}_2$ be the event that both conditions hold. The expected total regret $\mathbb{E}[R_T]$ can be bounded as:

$$\mathbb{E}[R_T] \leq \sum_{d=1}^{D} \mathbb{E}[T_d] + KL|\mathcal{A}^*| + \left( T - KL|\mathcal{A}^*| - \sum_{d=1}^{D} \mathbb{E}[T_d] \right) \cdot \mathbb{E}[\text{NSW}(\pi^*, s) - \text{NSW}(\hat{\pi}, s)]$$

$$\leq \sum_{d=1}^{D} \mathbb{E}[T_d] + KL|\mathcal{A}^*| + T \cdot \mathbb{E}[\text{NSW}(\pi^*, s) - \text{NSW}(\hat{\pi}, s) \mid \mathcal{E}] \cdot \mathbb{P}(\mathcal{E})$$

$$+ T \cdot \mathbb{P}(\mathcal{E}^c)$$

$$\leq \sum_{d=1}^{D} \mathbb{E}[T_d] + KL|\mathcal{A}^*| + T \cdot \mathbb{E}[\text{NSW}(\pi^*, s) - \text{NSW}(\hat{\pi}, s) \mid \mathcal{E}]$$

$$+ T \cdot \left( \mathbb{P}(\mathcal{E}_1^c) + \mathbb{P}(\mathcal{E}_2^c | \mathcal{E}_1) \right).$$

We now bound the term $\mathbb{E}[\text{NSW}(\pi^*, s) - \text{NSW}(\hat{\pi}, s) \mid \mathcal{E}]$.

$$\mathbb{E}[\text{NSW}(\pi^*, s) - \text{NSW}(\hat{\pi}, s) \mid \mathcal{E}] = \mathbb{E}[\text{NSW}(\pi^*, s) - \text{NSW}(\pi^*, \hat{s}) + \text{NSW}(\pi^*, \hat{s}) - \text{NSW}(\hat{\pi}, s) \mid \mathcal{E}]$$

$$\leq \mathbb{E}[|\text{NSW}(\pi^*, s) - \text{NSW}(\pi^*, \hat{s})| \mid \mathcal{E}] + \mathbb{E}[\text{NSW}(\hat{\pi}, \hat{s}) - \text{NSW}(\hat{\pi}, s)| \mid \mathcal{E}]$$

$$\leq 2D \cdot \sqrt{\frac{\log(DKT)}{L}},$$

where the first inequality is due to $\hat{\pi} \in \arg\max_\pi \text{NSW}(\pi, \hat{s})$ and the last inequality is due to Lemma A.1 and the definition of $\mathcal{E}_2$.

Next, we bound the terms related to the identification phase. The expected number of samples $\mathbb{E}[T_d]$ required for the DKWT algorithm to identify the Condorcet winner $a_d^*$ for user $d$ with an error probability of at most $\delta/D$ is bounded by (Haddenhorst et al., 2021):

$$\sum_{d=1}^{D} \mathbb{E}[T_d] \leq D \cdot O\left( \frac{K}{\Delta^2} \log(K/2) \left( \log\log\left(\frac{1}{\Delta}\right) + \log\left(\frac{D}{\delta}\right) \right) \right).$$

The probability of failure in Phase 1, $\mathbb{P}(\mathcal{E}_1^c)$, is the probability that at least one Condorcet winner is misidentified. By choosing the confidence parameter for the DKWT algorithm to be $\delta/D$ for each user $d$, the probability of error in identifying a single winner is $\mathbb{P}(\hat{a}_d^* \neq a_d^*) \leq \delta/D$. Using the union bound, the total probability of misidentification is:

$$\mathbb{P}(\mathcal{E}_1^c) = \mathbb{P}(\exists d : \hat{a}_d^* \neq a_d^*) \leq \sum_{d=1}^{D} \mathbb{P}(\hat{a}_d^* \neq a_d^*) \leq D \cdot \frac{\delta}{D} = \delta.$$

Next, we bound the probability of error in Phase 2, $\mathbb{P}(\mathcal{E}_2^c)$. Recall that $\mathcal{E}_2^c$ is the event that for some user $d$ and some arm $a$, the estimated score $\hat{s}_d(a)$ is significantly different from the true score $s_d(a)$. The score $\hat{s}_d(a)$ is an average of $L$ independent pairwise comparison outcomes, which are bounded in $[0, 1]$. By Hoeffding's inequality, for a single arm $a$ and user $d$:

$$\mathbb{P}\left( |\hat{s}_d(a) - s_d(a)| > \epsilon \right) \leq 2\exp(-2L\epsilon^2).$$

We choose the error bound $\epsilon = \sqrt{\frac{\log(DKT)}{L}}$ as defined in $\mathcal{E}_2$. Substituting this into the inequality:

$$\mathbb{P}\left( |\hat{s}_d(a) - s_d(a)| > \sqrt{\frac{\log(DKT)}{L}} \right) \leq 2\exp\left( -2L\frac{\log(DKT)}{L} \right) = 2\exp(-2\log(DKT)) = \frac{2}{(DKT)^2}.$$

Using the union bound over all $D$ users and $K$ arms, we get the total probability of failure for Phase 2:

$$\mathbb{P}(\mathcal{E}_2^c) = \mathbb{P}\left(\exists d \in [D], a \in [K] : |\hat{s}_d(a) - s_d(a)| > \sqrt{\frac{\log(DKT)}{L}}\right) \leq DK \cdot \frac{2}{(DKT)^2} = \frac{2}{DKT^2}.$$

Substituting the bounds derived in the previous steps into the expected regret equation:

$$\mathbb{E}[R_T] \leq \sum_{d=1}^{D} \mathbb{E}[T_d] + KL|\mathcal{A}^*| + T \cdot \left(2D \cdot \sqrt{\frac{\log(DKT)}{L}}\right) + T \cdot \left(\delta + \frac{2}{DKT^2}\right)$$

$$\leq D \cdot O\left(\frac{K}{\Delta^2} \log(K/2)\left(\log\log\left(\frac{1}{\Delta}\right) + \log\left(\frac{D}{\delta}\right)\right)\right) + KL|\mathcal{A}^*|$$

$$+ 2DT \cdot \sqrt{\frac{\log(DKT)}{L}} + T\delta + \frac{2}{DKT}.$$

Setting $\delta = \frac{K \log\left(\frac{K}{2}\right)}{2\hat{\Delta}T}$ and $L = \Theta(K^{-\frac{2}{3}}|\mathcal{A}^*|^{-\frac{2}{3}} D^{\frac{2}{3}} T^{\frac{2}{3}} \log(DKT)^{\frac{1}{3}})$ yields the theorem's statement. □

## A.3 Proof for Section 4.3

We first present a useful Lemma bounding the difference between the estimated scores and true scores.

**Lemma A.2.** *Let $n_{a,a^*}^t$ be the number of times arm $a$ was played against $a^*$ until time $t$. Then*

$$\mathbb{P}\left(d \in [D], a \in [K]|s_d(a) - \hat{s}_d^t(a)| > \sqrt{\frac{2\log(DKt)}{n_{a,a_d^*}^t}}\right) < \frac{2}{(DKt)^3}$$

*Proof.* Fix a time step $t \in [T]$. We bound the probability that the estimated score deviates significantly from the true score for any user $d$ and arm $a$ at this specific time step.

For any specific pair $(d, a)$, the number of samples collected, $n_{a,a_d^*}^t$, is a value $l$ in the range $\{1, \ldots, t\}$. We apply a union bound over all possible integer values $l \leq t$.

For a fixed number of samples $l$, let $\hat{s}_{d,l}(a)$ denote the average of $l$ independent Bernoulli variables. By Hoeffding's inequality, for the threshold $\epsilon_l = \sqrt{\frac{2\log(DKt)}{l}}$:

$$\mathbb{P}\left(|s_d(a) - \hat{s}_{d,l}(a)| > \sqrt{\frac{2\log(DKt)}{l}}\right) \leq 2\exp\left(-2l \cdot \frac{2\log(DKt)}{l}\right) = \frac{2}{(DKt)^4}.$$

By applying the Union bound, we get

$$\mathbb{P}\left(|s_d(a) - \hat{s}_d(a)| > \sqrt{\frac{2\log(DKt)\forall l \in [t], d \in [D], a \in [K]}{l}}\right) \leq \frac{2}{(DKt)^3}.$$

As $n_{a,a_d^*}^t \in [t]$, we get the statement of the lemma. □

**Proof of Theorem 4.3**

*Proof.* Let $\mathbb{E}[T_d]$ be the expected number of steps the DKWT algorithm identifies the Condorcet winner for user $d$ and $T_0 = \sum_{d=1}^{D} T_d$. Let $\tau_0 := (K-1)|\mathcal{A}^*|$ be the length of the deterministic warm-up phase, in which every pair $(a, a^*)$, $a^* \in \hat{\mathcal{A}}^*$, $a \neq a^*$, is played exactly once. The expected regret can then be written as

$$\mathbb{E}[R_T] \leq \sum_{d=1}^{D} \mathbb{E}[T_d] + \tau_0 + \mathbb{E}\left[\sum_{t=T_0+\tau_0+1}^{T} r_t\right]$$

The first term corresponds to the regret accumulated during the identification phase, the second term is the (trivially bounded, since NSW $\in [0,1]$) regret of the deterministic warm-up phase, contributing at most $\tau_0$, and the third term accounts for the main loop.

Similar to the proof of Theorem 4.2, the expected duration of the DKWT algorithm summing over all $D$ users is bounded by:

$$\mathbb{E}[T_0] \leq O\left(\frac{KD}{\Delta^2} \log\left(\frac{K}{2}\right) \left(\log\log\left(\frac{1}{\Delta}\right) + \log\left(\frac{D}{\delta}\right)\right)\right). \tag{9}$$

Let $\mathcal{E}_1$ be the event that all Condorcet winners are correctly identified in Phase 1. By the properties of DKWT, $\mathbb{P}(\mathcal{E}_1^c) \leq \delta$. Note that $\tau_0 = (K-1)|\mathcal{A}^*| = O(K|\mathcal{A}^*|)$ is a fixed quantity independent of $T$, $\delta$, and the exploration schedule $\epsilon_t$, since the warm-up phase is deterministic rather than governed by $\epsilon_t$.

Conditioned on $\mathcal{E}_1$, and on the warm-up phase having completed (which guarantees every pair $(a, a^*)$ has at least one sample before the main loop begins), the Condorcet winners are known. For $t > T_0 + \tau_0$, the algorithm explores with probability $\epsilon_t$ and exploits with probability $1 - \epsilon_t$. The expected regret at step $t$ is:

$$\mathbb{E}[r_t] \leq \epsilon_t \cdot 1 + (1 - \epsilon_t) \cdot \mathbb{E}[\mathrm{NSW}(\pi^*, s) - \mathrm{NSW}(\hat{\pi}_t, s)].$$

The first term accounts for the exploration. The second term is the regret from exploiting the estimated scores $\hat{s}$.

Let $\mathcal{E}_2^t$ be the event that the score estimates are concentrated around their true values at time step $t \in [T - T_0 - \tau_0]$, consistent with Lemma A.2. Let $t' = t - T_0 - \tau_0$,

$$\mathcal{E}_2^t = \left\{\forall d \in [D], a \in [K] : |s_d(a) - \hat{s}_d(a)| \leq \sqrt{\frac{2\log(DKt')}{n_{a,a_d^*}^t}}\right\}. \tag{10}$$

By Lemma A.2, the probability of the complement event is bounded by $\mathbb{P}(\mathcal{E}_2^{t^c}) \leq \frac{2}{(DKt)^3}$.

Next, we analyze the number of exploration steps for a fixed time $t$. Let $N^t = \sum_{\tau=T_0+\tau_0+1}^{t} I_\tau$ be the actual number of main-loop exploration steps up to time $t$ (not counting the warm-up phase, which is accounted for separately). Note that $\mathbb{E}[N^t] = \sum_{\tau=T_0+\tau_0+1}^{t} \epsilon_\tau \geq (t - T_0 - \tau_0)\epsilon_t$ as $\epsilon_\tau$ is monotonically decreasing for the chosen value of $\epsilon$.

Let $0 < \gamma < 1$ be a constant such that $\epsilon_t \geq \frac{1}{\gamma(t-T_0-\tau_0)^{\frac{1}{3}}}$. We define the event $\mathcal{E}_3^t$ as follows

$$\mathcal{E}_3^t = \left\{N^t \geq \mathbb{E}[N^t] - \gamma\mathbb{E}[N^t]\right\}. \tag{11}$$

We bound the probability of the complement event $\mathcal{E}_3^{t^c}$ using the Hoeffding inequality. Since $I_\tau \in [0,1]$ are independent random variables and the sum consists of at most $t - T_0 - \tau_0$ terms:

$$\mathbb{P}(\mathcal{E}_3^{t^c}) \leq \exp\left(-\frac{2\gamma^2(\mathbb{E}[N^t])^2}{t - T_0 - \tau_0}\right) \leq \exp\left(-\frac{2\gamma^2((t - T_0 - \tau_0)\epsilon_t)^2}{t - T_0 - \tau_0}\right) = \exp\left(-2\gamma^2(t - T_0 - \tau_0)(\epsilon_t)^2\right)$$

$$\leq \exp\left(-2(t - T_0 - \tau_0)^{\frac{1}{3}}\right) \leq \exp\left(-\log(t - T_0 - \tau_0)\right) = \frac{1}{t - T_0 - \tau_0}.$$

Because the warm-up phase already guarantees each pair $(a, a^*)$ one sample, the round-robin schedule in the main loop only needs to add further coverage, and the standard floor identity $\lfloor x \rfloor + 1 \geq x$ (for $x \geq 0$) gives an exact bound: given $\mathcal{E}_1$ and $\mathcal{E}_3^t$,

$$n_{a,a^*}^t \geq 1 + \left\lfloor\frac{N^t}{(K-1)|\mathcal{A}^*|}\right\rfloor \geq \frac{N^t}{(K-1)|\mathcal{A}^*|} \geq \frac{(1-\gamma)(t - T_0 - \tau_0)\epsilon_t}{(K-1)|\mathcal{A}^*|}.$$

Hence, event $\mathcal{E}_2^t$ can be rewritten as:

$$\mathcal{E}_2^t = \left\{\forall d \in [D], a \in [K] : |s_d(a) - \hat{s}_d^t(a)| \leq \sqrt{\frac{2(K-1)|\mathcal{A}^*|\log(DKt')}{(1-\gamma)(t - T_0 - \tau_0)\epsilon_t}}\right\}. \tag{12}$$

We can now decompose the expected regret in the exploitation phase conditioned on $\mathcal{E} = \mathcal{E}_1 \cap \mathcal{E}_2^t \cap \mathcal{E}_3^t$:

$$\mathbb{E}[r_t] \le \epsilon_t + (1 - \epsilon_t) \left( \mathbb{E}[\mathrm{NSW}(\pi^*, s) - \mathrm{NSW}(\hat{\pi}_t, s) \mid \mathcal{E}] + \mathbb{P}(\mathcal{E}^c) \right)$$
$$\le \epsilon_t + \mathbb{E}[\mathrm{NSW}(\pi^*, s) - \mathrm{NSW}(\hat{\pi}_t, s) \mid \mathcal{E}] + \delta + \frac{2}{(DKT)^3} + \frac{1}{t - T_0 - \tau_0}.$$

We further decompose $\mathbb{E}[\mathrm{NSW}(\pi^*, s) - \mathrm{NSW}(\hat{\pi}_t, s) \mid \mathcal{E}]$.

$$\mathbb{E}[\mathrm{NSW}(\pi^*, s) - \mathrm{NSW}(\hat{\pi}_t, s) \mid \mathcal{E}] = \mathbb{E}[\mathrm{NSW}(\pi^*, s) - \mathrm{NSW}(\pi^*, \hat{s}^t) + \mathrm{NSW}(\pi^*, \hat{s}^t) - \mathrm{NSW}(\hat{\pi}_t, s) \mid \mathcal{E}]$$
$$\le \mathbb{E}[\mathrm{NSW}(\pi^*, s) - \mathrm{NSW}(\pi^*, \hat{s}^t) + \mathrm{NSW}(\hat{\pi}_t, \hat{s}) - \mathrm{NSW}(\hat{\pi}_t, s) \mid \mathcal{E}]$$
$$\le 2D\sqrt{\frac{2(K-1)|\mathcal{A}^*| \log(DKt')}{(1-\gamma)(t - T_0 - \tau_0)\epsilon_t}},$$

where the first inequality is due to $\hat{\pi}_t \in \arg\max_{\pi \in \Delta_K} \mathrm{NSW}(\pi, \hat{s}^t)$ and the last inequality due to Lemma A.2.

Let $T' = T - T_0 - \tau_0$, substituting these bounds back into the regret decomposition and summing over $t = T_0 + \tau_0 + 1, \ldots, T$, and adding the warm-up contribution of at most $\tau_0$, we obtain the bound for the total expected regret:

$$\mathbb{E}[R_T] \le O\left( \frac{KD}{\Delta^2} \log\left(\frac{K}{2}\right) \left( \log\log\left(\frac{1}{\Delta}\right) + \log\left(\frac{D}{\delta}\right) \right) \right) + \tau_0$$
$$+ \sum_{t'=1}^{T'} \left( \epsilon_{t'+T_0+\tau_0} + 2D\sqrt{\frac{2(K-1)|\mathcal{A}^*| \log(DKT)}{(1-\gamma)t'\epsilon_{t'+T_0+\tau_0}}} + \delta + \frac{2}{(DKT)^3} + \frac{1}{t'} \right)$$
$$\le O\left( \frac{KD}{\Delta^2} \log\left(\frac{K}{2}\right) \left( \log\log\left(\frac{1}{\Delta}\right) + \log\left(\frac{D}{\delta}\right) \right) \right) + O(K|\mathcal{A}^*|)$$
$$+ \sum_{t'=1}^{T'} \left( \epsilon_{t'+T_0+\tau_0} + 2D\sqrt{\frac{2(K-1)|\mathcal{A}^*| \log(DKT)}{(1-\gamma)t'\epsilon_{t'+T_0+\tau_0}}} + \frac{2}{(DKT)^3} \right) + \delta T' + 1 + \log(T')$$

The added warm-up term $\tau_0 = O(K|\mathcal{A}^*|)$ is independent of $T$ and is dominated by the $T^{2/3}$ term below.

Setting $\delta = \frac{K \log(\frac{K}{2})}{2\hat{\Delta}T}$ and $\epsilon_t = \Theta(D^{\frac{2}{3}} K^{\frac{1}{3}} |\mathcal{A}^*|^{\frac{1}{3}} (t - T_0 - \tau_0)^{-\frac{1}{3}} \log^{\frac{1}{3}}(DKt'))$ for $t > T_0 + \tau_0$ yields the theorem's statement. $\square$

## B  Missing Algorithms

In this section, we present the detailed pseudocode for the algorithms employed during the Condorcet winner identification phase. Specifically, we detail the **DKW-Compare** subroutine (Algorithm 3) and the **Modified DKWT** algorithm (Algorithm 4). These methods are adaptations of the original Dvoretzky-Kiefer-Wolfowitz Tournament (DKWT) algorithm proposed by Haddenhorst et al. (2021), modified here to handle multiple heterogeneous users in parallel.

---

**Algorithm 3** DKW-Compare

---

**Require:** Arms $i, j$, Calling User $d$, Confidence $\delta'$, sample access to $\mathcal{P}$, Global candidate sets $\mathcal{S}$, Global round trackers $R$

1: Initialize round $r \leftarrow R(i, j)$.
2: **while** $\{i, j\} \subseteq S_d$ **do**
3:     Set confidence width $h_r \leftarrow 2^{-(r+1)}$.
4:     Set adjusted confidence $\delta_r \leftarrow \frac{6\delta'}{\pi^2 r^2}$.
5:     Calculate samples $N_r \leftarrow \lceil 8 \log(4/\delta_r)/h_r^2 \rceil$.
6:     Play duel $(i, j)$ for $N_r$ times and record outcomes for all users.
7:     Let $U_{i,j} \leftarrow \{d' \in [D] \mid \{i, j\} \subseteq S_{d'}\}$ be the set of all users still comparing $i$ and $j$.
8:     **for** $d' \in U_{i,j}$ **do**
9:         Compute empirical probability $\hat{p}_{d'} = \hat{\mathcal{P}}_{d'}(i, j)$.
10:        **if** $|\hat{p}_{d'} - 0.5| > 0.5h_r$ **then**
11:           **if** $\hat{p}_{d'} > 0.5$ **then**
12:             $S_{d'} \leftarrow S_{d'} \setminus \{j\}$
13:           **else**
14:             $S_{d'} \leftarrow S_{d'} \setminus \{i\}$
15:           **end if**
16:        **end if**
17:     **end for**
18:     $r \leftarrow r + 1$.
19:     $R(i, j) \leftarrow r$
20: **end while**

---

**Algorithm 4** Modified DKWT for Multiple Users

---

**Require:** Set of Arms $[K]$, Set of Users $[D]$, sample access to $\mathcal{P}$, Confidence $\delta$.
**Ensure:** Set of estimated Condorcet winners $\hat{\mathcal{A}}^*$.

1: Initialize $\hat{\mathcal{A}}^* \leftarrow \emptyset$.
2: Initialize global candidate sets $S_d \leftarrow [K]$ for all $d \in [D]$.
3: Initialize global round trackers $R(i, j) \leftarrow 1$ for all distinct pairs $i, j \in [K]$
4: **while** $\exists d \in [D]$ such that $|S_d| > 1$ **do**
5:     Choose a user $d$ where $|S_d| > 1$.
6:     Select a distinct pair $i, j \in S_d$.
7:     **DKW-Compare**$(i, j, d, \frac{\delta}{K}, \mathcal{P})$.
8: **end while**
9: **for** $d \in [D]$ **do**
10:     Let $a_d^*$ be the single element in $S_d$.
11:     Add $a_d^*$ to $\hat{\mathcal{A}}^*$.
12: **end for** **return** $\hat{\mathcal{A}}^*$

---

