# OpenReview forum: "Multi-User Dueling Bandits: A Fair Approach using Nash Social Welfare"
_TMLR — Under review for TMLR_

### Review · Reviewer_wQwQ · 2026-07-07

**Summary Of Contributions:**

**Summary**

The paper introduces fairness into dueling bandits with heterogeneous users, where fairness is captured through the NSW (Nash Social Welfare) objective. It defines user utilities relative to each user’s Condorcet winner, thereby providing a way to convert pairwise preference feedback into welfare-based scores. The paper then proves a regret lower bound that characterizes the intrinsic cost of fairness in this setting, and proposes Fair-ETC and Fair-$\epsilon$-Greedy algorithms with matching $T^{2/3}$-type regret guarantees up to logarithmic factors. Finally, experiments on synthetic and Sushi preference data show that the proposed methods improve fairness metrics such as minimum welfare and the Gini coefficient compared with utilitarian baselines.

---

**Strengths**

1. The paper studies a well-motivated combination of ideas: Nash Social Welfare fairness and relative/preference-based feedback. If one only optimizes average preference, the preferences of minority users or minority groups may be sacrificed over the long run, which is an issue of clear interest to the community.

2. The formulation is relatively clean, the paper is easy to follow, the algorithms are simple and direct, and the intuition behind the upper and lower bounds is also explained clearly.

---

**Weaknesses**

1. The regret bounds match in terms of the dependence on $T$, but it is less clear whether the dependence on other problem parameters, such as $K$, $D$, $|A^*|$, and the preference gap $\Delta$, is tight or can be improved.

2. The use of the product-form Nash Social Welfare objective raises some scaling and interpretability issues. When $D$ is large, the product can become extremely small and highly sensitive to any user whose utility is close to zero. The lower-bound construction also seems to exploit this sensitivity. It would be helpful for the paper to discuss whether it is desirable in practical applications, and to clarify the relationship between the theoretical product objective and the geometric-mean-style fairness metrics reported in the experiments.

**Audience:**

Yes

**Audience Explanation:**

Yes. The paper connects heterogeneous user preferences with NSW in the dueling bandits and provides both theoretical regret results and empirical evidence. Although the setting is somewhat specialized, the question of how to fairly aggregate diverse preferences is relevant to broader machine learning research.

**Broader Impact Concerns:**

There is no concern on the ethical implications since this is a theoretical work with synthetic experiments.

**Claims And Evidence:**

Yes

**Claims Explanation:**

1. The claims are mostly supported. There are some minor proof issues that should be fixed.

2. The relationship between the theoretical product-form NSW objective and the experimental metrics could be clarified.

**Requested Changes:**

1. The authors should clarify the relationship between the theoretical product-form NSW objective and the experimental metrics. The proof issues should be fixed.

2. It would also be helpful to add more discussion on the tightness of the dependence on the problem parameters.

---

### Review · Reviewer_6fNa · 2026-07-07

**Summary Of Contributions:**

This paper investigates an interesting problem of fairness in multi-user dueling bandits. Rather than treating the problem as a standard multi-armed bandit with scalar feedback representing the average utility across users, the authors consider an online learning setting where the learner observes only pairwise preferences between two arms. Fairness across heterogeneous users is captured through the Nash Social Welfare (NSW) objective. The authors derive a regret lower bound of $\Omega(T^{2/3} min(K,D)^{1/3})$ and propose two algorithms, which includes Fair-ETC (Explore-then-Commit) and Fair-$\epsilon$-Greedy.

Strenths:
1. The paper is timely, well-written, and easy to follow. The related work section is thorough and covers most of the recent literature on fairness in bandits and RL.
2. Fairness is an important problem in many real-world applications, and extending it to the dueling bandit setting is both natural and well motivated. I particularly like the preference-based formulation, where the learner observes only relative feedback instead of absolute rewards, making it conceptually closer to RLHF and other learning settings from human feedback.
3. The paper is technically solid. The regret lower bound, in my opinion, is the main contribution of the paper, and together with the two algorithms and their extensive empirical validations, makes this an even stronger paper.

Overall, I find the paper to be in the right direction and a meaningful contribution. Below I raise some concerns that, if addressed, would further strengthen the paper.
Weaknesses:
1. Although the authors acknowledge it, the assumption that every user has a unique Condorcet winner is quite strong, and the paper relies heavily on it. While this assumption simplifies the analysis, it also limits the applicability of the proposed methods. I encourage the authors to discuss in more detail how the proposed methods would behave when this assumption is violated.
2. The authors defined utility as, $s_d (i) = 2P_{d, i, a*_d}$, which is actually a rescaled probability that arm $i$ beats user $d$'s own Condorcet winner. Once the Condorcet winners are identified, this actually defines a scalar utility for each user. At that point, the formulation appears conceptually closer to that of Hossain et al. (2021), and it is not entirely clear what additional role the dueling formulation plays beyond estimating these utilities.
3. While the related work discusses several welfare functions used in fair RL, the authors only use NSW. I agree that NSW has many desirable properties, but it is unclear whether the analysis or algorithms extend to other welfare objectives, such as $\alpha$-fairness, the Generalized Gini Function, or generalized $p$-means.
4. The proposed algorithms are also restrictive in their own way. Fair-ETC commits after a fixed exploration budget and cannot recover from an inaccurate estimate, whereas Fair-$\epsilon$-Greedy performs NSW optimization at each round during learning, which may become computationally expensive. The trade-off between these two methods is not fully discussed in the paper and should be included.

**Audience:**

Yes

**Audience Explanation:**

Fairness in bandits, dueling bandits, or in preference-based online learning is directly relevant to RL from human feedback and other decision-making systems that must serve multiple users and is currently an active area of research. Therefore, I believe the topic will be of interest to the community. That said, the reliance on strong structural assumptions, particularly the unique Condorcet winner assumption and the exclusive use of NSW, may limit the practical applicability of the proposed methods.

**Broader Impact Concerns:**

None foreseen.

**Claims And Evidence:**

Yes

**Claims Explanation:**

Most of the claims are supported by the theoretical analysis and empirical evaluation. The preference-based formulation naturally fits the dueling bandit setting, and the fairness objective is clearly defined through NSW. My concerns are mostly related to the scope of the work and the practical applicability of the proposed methods rather than to the correctness.

**Requested Changes:**

1. Can the authors discuss whether the analysis and methods extend to other welfare functions? Authors should ideally provide at least a discussion or experiment beyond NSW. For example, GGF has been studied in fair bandits[1] and MORL. It would also be useful and interesting to see if the lower bound is specific to NSW or applies to other classes of welfare functions as well.
2. Why is the Condorcet winner the appropriate reference point for defining user utilities? What will happen in settings where no Condorcet winner exists? Some discussion of it would substantially broaden the scope of the paper.
3. Since Fair-ETC commits after exploration, an incorrect estimate cannot be corrected later. Can the authors quantify how sensitive the algorithm is to the exploration budget $L$, and how frequently early commitment leads to suboptimal performance?
4. Fair-ETC (Explore-then-Commit) and Fair-$\epsilon$-Greedy seem to offer diffenet trade-off. For example, Fair-ETC achieves competitive cumulative regret and NSW, while Fair-$\epsilon$-Greedy performs slightly better on minimum welfare and Gini coeff. Could the authors provide clearer guidance on when each algorithm should be preferred, including a comparison of their computational costs?
5. Related to my previous point, in Table 1, the Uniform-Over-Users achieves both the lowest Gini coefficient and the highest minimum welfare, while the proposed methods achieve the best NSW and cumulative regret. Similarly, the Utilitarian achieves the highest Utilitarian Welfare and comparable NSW. This is interesting because the price for fairness seems to be too high for both proposed methods, while utilitarian being the best in terms of rewards is already achieving a comparable NSW. Could the authors quantify the price of fairness more explicitly? In particular, can they confirm that the proposed methods have fully converged and provide a systematic measure of the fairness-efficiency trade-off rather than relying on separate metrics?


[1]  Busa-Fekete, R., et al. "Multi-objective bandits: Optimizing the generalized Gini index." ICML, 2017.

---

### Review · Reviewer_Qrcz · 2026-07-08

**Summary Of Contributions:**

This paper studies fairness in dueling bandits with multiple heterogeneous users. Each user has a distinct pairwise preference matrix and a unique Condorcet winner. Since dueling feedback is relative rather than cardinal, the paper defines an arm’s utility for a user by comparing that arm against the user’s own Condorcet winner, namely $s_d(i)=2P_d(i,a_d^*)$. The learning objective is then to maximize the Nash Social Welfare (NSW), defined as the product of users’ expected utilities under a mixed policy.

The paper’s main claimed contributions are: (i) a fair multi-user dueling-bandit formulation based on user-specific Condorcet-reference scores; (ii) an information-theoretic lower bound of $\Omega(T^{2/3}\min(K,D)^{1/3})$, arguing that fairness makes exploration more costly than in standard Condorcet dueling bandits; (iii) two algorithms, Fair-ETC and Fair-$\epsilon$-Greedy, which first identify user-specific Condorcet winners and then estimate Condorcet-relative scores before optimizing empirical NSW; and (iv) experiments on synthetic instances and the Sushi preference dataset comparing the proposed methods with utilitarian and uniform-over-users baselines.

The main strengths are the timely connection between fairness, preference feedback, and dueling bandits; the clean Condorcet-relative utility construction; and the lower-bound intuition that fair learning may require socially costly comparisons. The empirical section also gives useful evidence that NSW-based policies can improve minimum welfare and inequality metrics relative to utilitarian baselines.
The main weaknesses are technical. Several proof steps, theorem statements, and pseudocode details are currently inconsistent. These issues affect the claimed regret upper bounds, the lower-bound proof, and the precise interpretation of the experimental evidence. Most of these issues appear fixable, but they are central enough that the current version does not yet provide fully convincing support for the main theoretical claims.

**Audience:**

Yes

**Audience Explanation:**

The paper addresses a relevant problem at the intersection of preference learning, bandit algorithms, and algorithmic fairness. Dueling bandits are a natural abstraction for learning from pairwise human feedback, and the multi-user setting with heterogeneous preferences is well motivated by recommendation, ranking, and preference-alignment applications. The proposed use of NSW is also natural for balancing efficiency and fairness across users.
The most interesting aspect of the paper is the claim that fair learning in dueling bandits can be fundamentally harder than standard Condorcet dueling bandits, with a $T^{2/3}$ lower-bound phenomenon arising from socially costly exploration. If corrected and presented rigorously, this result would likely be of interest to researchers working on online learning, preference-based RL, RLHF-style feedback models, and fairness in sequential decision-making.

**Broader Impact Concerns:**

I do not see a broader-impact concern that would by itself preclude publication. However, because the paper is explicitly about fairness in systems trained from human preference feedback, the broader-impact discussion should be more explicit if it is not already included.

**Claims And Evidence:**

Yes

**Claims Explanation:**

The high-level formulation is plausible and the empirical trends are suggestive, but the current manuscript does not yet support all of its main claims with sufficiently accurate and clear evidence. In particular, the proof of the upper bounds contains inconsistencies in the DKWT confidence term, the Lipschitz bound used for NSW, and the concentration argument for the round-robin exploration schedule. The pseudocode also does not consistently match the score-estimation problem: the required score $s_d(a)=2P_d(a,a_d^*)$ can only be directly estimated from comparisons against the relevant Condorcet winner, while Algorithm 2 appears to update score estimates using arbitrary exploitation duels.

There is also a change-of-measure issue in the lower-bound proof: the proof writes the KL divergence in one direction but later bounds it using action probabilities under the other instance. This should either be justified carefully or corrected by reversing the KL direction. Finally, the theoretical NSW objective is the product of expected user utilities, while the experiments report a geometric mean of cumulative utilities. These objectives have the same maximizers only under fixed-dimensional monotone transformations, but regret values and scaling are not the same.

These issues may be repairable, and I do not view them as invalidating the direction of the work. However, they prevent the present version from providing convincing evidence for the stated regret guarantees and for the quantitative interpretation of the empirical regret curves.

**Requested Changes:**

**Critical Concerns**

- Correct the DKWT logarithmic factor in Theorems 4.2 and 4.3. In the proof, the DKWT identification cost is bounded by $O\left( \frac{KD}{\Delta^2}\log(K/2) \left[\log\log(1/\Delta)+\log(D/\delta)\right] \right). $ With $\delta=\frac{K\log(K/2)}{2\hat{\Delta}T}, $ this gives $\log(D/\delta)\log\left(\frac{2D\hat{\Delta}T}{K\log(K/2)}\right).$
However, the theorem statements use
$
\log\left(\frac{2\hat{\Delta}T}{KD\log(K/2)}\right),
$
which places $D$ in the denominator rather than the numerator. The authors should correct either the theorem statements or the proof. They should also state the parameter regime under which the chosen $\delta$ is a valid confidence parameter.

- State and prove the Lipschitz bound actually used in the regret proof. Lemma A.1 states
   $
   |\mathrm{NSW}(\pi,s^1)-\mathrm{NSW}(\pi,s^2)|
   \le
   \sum_{d=1}^D\sum_{i=1}^K |s^1_{d,i}-s^2_{d,i}|.
   $
   Under event $E_2$, this lemma only gives an $O(DK\eta)$ bound when $\eta=\sqrt{\frac{\log(DKT)}{L}}.$ The proof of Theorem 4.2 instead uses $2D\eta$, without the factor $K$. This can be obtained from the sharper weighted bound
   $
   |\mathrm{NSW}(\pi,s^1)-\mathrm{NSW}(\pi,s^2)|
   \le
   \sum_{d=1}^D\sum_{i=1}^K \pi_i |s^1_{d,i}-s^2_{d,i}|,
   $
   followed by $\sum_i\pi_i=1$. The authors should state and prove this weighted lemma, or otherwise adjust the regret bound to match the lemma as written.

- Align the algorithms with the Condorcet-relative score estimator.
   The target score is $s_d(a)=2P_d(a,a_d^\*),$
   so direct estimation requires comparisons of $a$ against the relevant user-specific Condorcet winner $a_d^\*$. Algorithm 2, line 9, says that arbitrary exploitation duels $(a_t,a_t')$ update $\hat{s}_t$. Such duels generally do not estimate $P_d(a,a_d^\*)$, unless one of the two arms is $a_d^\*$. The authors should either remove this update, restrict it to valid Condorcet-winner comparisons, or provide a different estimator and analysis showing how arbitrary pairwise comparisons are converted into Condorcet-relative scores.

- Fix the round-robin sample-count argument in Theorem 4.3.
   The proof claims that, after $N^t$ exploration steps,
   $
   n^t_{a,a_d^\*}\ge \frac{N^t}{K|A^\*|}.
   $
   A round-robin schedule over pairs $(a,a^\*)$ gives at best
   $
   n^t_{a,a^\*}\ge
   \left\lfloor
   \frac{N^t}{(K-1)|A^\*|}
   \right\rfloor
   $
   for pairs that have entered the cycle, and some pairs have zero samples before a full cycle is completed. This affects Equation (12) and the concentration bound based on $n^t_{a,a_d^\*}$. The proof should be revised, for example by adding an initial full-cycle exploration phase, using a floor/subtraction term, and explicitly handling early rounds with zero samples.

- Correct the change-of-measure argument in the lower bound.
   Equation (7) uses $D_{\mathrm{KL}}(I^m,I^0)$. For adaptive bandit experiments, the chain-rule decomposition of this KL divergence takes expectation under $I^m$. Later, however, the proof bounds the KL using probabilities under $I^0$, such as $\Pr_{I^0}(i_t=a_d^\*)$. This is not justified as written. A simple possible fix is to reverse the KL direction and use $D_{\mathrm{KL}}(I^0,I^m)$, since Pinsker’s inequality can be applied with either KL direction. Alternatively, the authors should provide a rigorous change-of-measure argument explaining why the probabilities under $I^0$ are valid in the displayed bound.

- Clarify the vector-feedback notation in Algorithm 1.
   Algorithm 1, line 5, uses $I(y_t=1)$, but the feedback $y_t\in\\{0,1\\}^D$ is a vector. The update should use $I(y_t[d]=1)$ for user $d$. The notation also alternates between estimating $\hat P_d(a,a^\*)$ for every $a^\*\in\hat A^\*$ and estimating only $\hat P_d(a,a_d^\*)$, which is the quantity needed by the NSW objective. The authors should make the indexing explicit and consistent.

- Reconcile the theoretical NSW objective with the experimental metric.
   The theory defines $\mathrm{NSW}(\pi,s)=\prod\_{d=1}^{D}\mathbb{E}\_{a\sim\pi}[s\_d(a)]$
, while the experiments report the geometric mean  $\left(\prod\_{d=1}^{D}\hat{u}\_{d}(T)\right)^{1/D}$
 of cumulative user utilities. These two quantities are related but not identical; in particular, regret values and scaling differ, especially across different $D$. The authors should state exactly which objective is used in each plot and table, and should avoid interpreting geometric-mean cumulative welfare as the same quantity as the product-based instantaneous NSW regret unless the normalization and monotone transformation are made explicit.



**Minor Concerns**

- The main text says a single duel gives feedback for all users, but Algorithm 3 says to record outcomes for the calling user $d$ only. Since the algorithm then updates candidate sets for multiple users $d'$, the pseudocode should explicitly record and use the full feedback vector for all active users.

- The algorithms require solving an empirical NSW maximization problem over the simplex. The paper mentions Frank-Wolfe in the experiments, but the theoretical algorithm statements use an exact argmax. It would help to specify whether the objective is optimized as $\sum_d \log u_d(\pi)$, how zero estimated utilities are handled, what approximation error is allowed, and how such error would affect regret.

- The empirical section would be more convincing with sensitivity analysis for the exploration-length and $\epsilon_t$ scaling constants, reporting of Frank-Wolfe stopping criteria, and additional plots showing dependence on $D$, $K$, $\Delta$, and $|A^\*|$. Since the theoretical constants are large and the experiments use scaled-down exploration, it would be useful to show that the qualitative conclusions are robust to these implementation choices.

- The preprocessing projects near-tie pairwise probabilities to the nearest boundary of $\\{0.4,0.6\\}$, but this alone does not necessarily guarantee the existence of a unique Condorcet winner in every cluster. The authors should explicitly report the Condorcet winner of each cluster after preprocessing, the minimum gap, and whether any clusters required additional adjustments.

- The proposed utility is simple and analyzable, but it depends strongly on the existence and identification of user-specific Condorcet winners. It would strengthen the paper to discuss whether the utility is robust to near ties, cyclic preferences, noisy user clusters, and alternative reference scores such as Borda or Copeland scores.